# Bacterioplankton Communities in Dissolved Organic Carbon-Rich Amazonian Black Water

François-Étienne Sylvain,[a] Sidki Bouslama,[a] Aleicia Holland,[b] Nicolas Leroux,[a] Pierre-Luc Mercier,[a] Adalberto Luis Val,[c] Nicolas Derome[a]

[a]Institut de Biologie Intégrative et des Systèmes, Université Laval, Québec, Quebec, Canada

[b]La Trobe University, School of Life Science, Department of Ecology, Environment and Evolution, Centre for Freshwater Ecosystems, Albury/Wodonga Campus, Victoria, Australia

[c]Instituto Nacional de Pesquisas da Amazônia, Laboratório de Ecofisiologia e Evolução Molecular, Manaus, Brazil

**ABSTRACT** The Amazon River basin sustains dramatic hydrochemical gradients defined by three water types: white, clear, and black waters. In black water, important loads of allochthonous humic dissolved organic matter (DOM) result from the bacterioplankton degradation of plant lignin. However, the bacterial taxa involved in this process remain unknown, since Amazonian bacterioplankton has been poorly studied. Its characterization could lead to a better understanding of the carbon cycle in one of the Earth's most productive hydrological systems. Our study characterized the taxonomic structure and functions of Amazonian bacterioplankton to better understand the interplay between this community and humic DOM. We conducted a field sampling campaign comprising 15 sites distributed across the three main Amazonian water types (representing a gradient of humic DOM), and a 16S rRNA metabarcoding analysis based on bacterioplankton DNA and RNA extracts. Bacterioplankton functions were inferred using 16S rRNA data in combination with a tailored functional database from 90 Amazonian basin shotgun metagenomes from the literature. We discovered that the relative abundances of fluorescent DOM fractions (humic-, fulvic-, and protein-like) were major drivers of bacterioplankton structure. We identified 36 genera for which the relative abundance was significantly correlated with humic DOM. The strongest correlations were found in the *Polynucleobacter*, *Methylobacterium*, and *Acinetobacter* genera, three low abundant but omnipresent taxa that possessed several genes involved in the main steps of the β-aryl ether enzymatic degradation pathway of diaryl humic DOM residues. Overall, this study identified key taxa with DOM degradation genomic potential, the involvement of which in allochthonous Amazonian carbon transformation and sequestration merits further investigation.

**IMPORTANCE** The Amazon basin discharge carries an important load of terrestrially derived dissolved organic matter (DOM) to the ocean. The bacterioplankton from this basin potentially plays important roles in transforming this allochthonous carbon, which has consequences on marine primary productivity and global carbon sequestration. However, the structure and function of Amazonian bacterioplanktonic communities remain poorly studied, and their interactions with DOM are unresolved. In this study, we (i) sampled bacterioplankton in all the main Amazon tributaries, (ii) combined information from the taxonomic structure and functional repertory of Amazonian bacterioplankton communities to understand their dynamics, (iii) identified the main physicochemical parameters shaping bacterioplanktonic communities among a set of >30 measured environmental parameters, and (iv) characterized how bacterioplankton structure varies according to the relative abundance of humic compounds, a by-product from the bacterial degradation process of allochthonous DOM.

Address correspondence to François-Étienne Sylvain, francois-etienne.sylvain.1@ulaval.ca.

The authors declare no conflict of interest.

**KEYWORDS** *Acinetobacter*, *Methylobacterium*, *Polynucleobacter*, bacterioplankton, carbon cycle, dissolved organic carbon, dissolved organic matter, humic acids, microbiome

The Amazon basin occupies almost 38% of continental South America (1) and holds 12% to 20% of the planet's liquid freshwater (2). Its discharge carries a significant load of terrestrially derived nutrients to the ocean, which have global consequences on marine primary productivity and global carbon sequestration (3, 4). The Amazon River basin sustains dramatic hydrochemical and ecological gradients that impose physiological constraints upon its aquatic communities (5–9). Its three major tributaries, the Rio Solimões, Rio Negro, and Rio Tapajos, represent distinct water "types" or "colors" that harbor contrasting physicochemical profiles (10). The white water from the Rio Solimões has an Andean origin, is eutrophic (nutrient- and ion-rich), turbid, and has a circumneutral pH (10–13). The crystalline "clear water" from the Rio Tapajos has a circumneutral pH, low conductivity, and a reduced amount of suspended material associated with its pre-Cambrian rock origin draining the Brazilian shield. Last, the black water of the Rio Negro stems from the craton born drainage of the Guyana shield (14) and largely contrasts with the aforementioned tributaries; it is oligotrophic (nutrient- and ion-poor) and contains a high quantity of dissolved organic matter (DOM), typically 8 to 12 mg C/liter (15). Black water DOM is enriched in chromophoric dissolved organic matter (CDOM), a fraction of DOM that absorbs light. While the measure of fluorescent CDOM (or fluorescent dissolved organic matter [FDOM]) can be used to characterize the nature of DOM in a system, FDOM only represents a fraction of the total DOM, varying between <10% and 70% depending on the environment (16, 17). FDOM from the Rio Negro has a distinctive allochthonous origin, in comparison with FDOM from the Rio Solimões or Rio Tapajos (18).

Overall, the Amazon basin has very low rates of phytoplankton production (19), suggesting that terrestrial allochthonous DOM is an important carbon source for bacterial growth (20, 21). This is especially true in the Rio Negro's black water, where most of the DOM is a by-product of the lignin degradation process associated with plant decomposition on the riverbed, fueled by the important release of plant material during the seasonal forest flooding (10, 22, 23). The DOM from the Rio Negro is characterized by a complex mixture of humic aromatic compounds, which mostly originate from the first step of the lignin degradation process, the microbial oxidation of lignin-derived compounds (Fig. 1) (22, 24). While fungi have been particularly studied for their involvement in this process in the past, some studies are starting to unravel the lignin and DOM degradation machinery of bacterioplankton (25, 26).

Despite its relevance for global-scale elemental cycling and primary production processes, there is a limited understanding of the taxonomical and functional structure of the Amazon River bacterioplankton. A few studies have focused on these bacterial communities; however, most of them did not sample in different water types (21–29). Several of the aforementioned studies also included a limited number of habitats sampled (e.g., only one site in reference 23) or were focused on the dynamics of bacterioplankton in the plume downstream of the Amazonian River (25–27) rather than the communities in upstream black water systems *per se*. A previous study (21) found that the genera *Ramlibacter*, *Planktophila*, *Methylopumilus*, *Limnohabitans*, and *Polynucleobacter* were enriched in DOM degradation pathways along the Amazon River; however, the abundance of these genera in accordance with humic DOM and whether they are transcriptionally active or not remain unknown.

In this study, we aimed to identify the most important environmental variables shaping the Amazonian bacterioplankton community structure and inferred functional profile. Given that bacterioplankton communities have been shown to be involved in the transformation of DOM (30–32), we hypothesized that the optical characteristics of DOM in particular FDOM, such as humic-like materials, would be one of the most important drivers of bacterioplankton communities. In particular, we expected that the relative abundance of humic FDOM would significantly affect bacterioplankton communities as

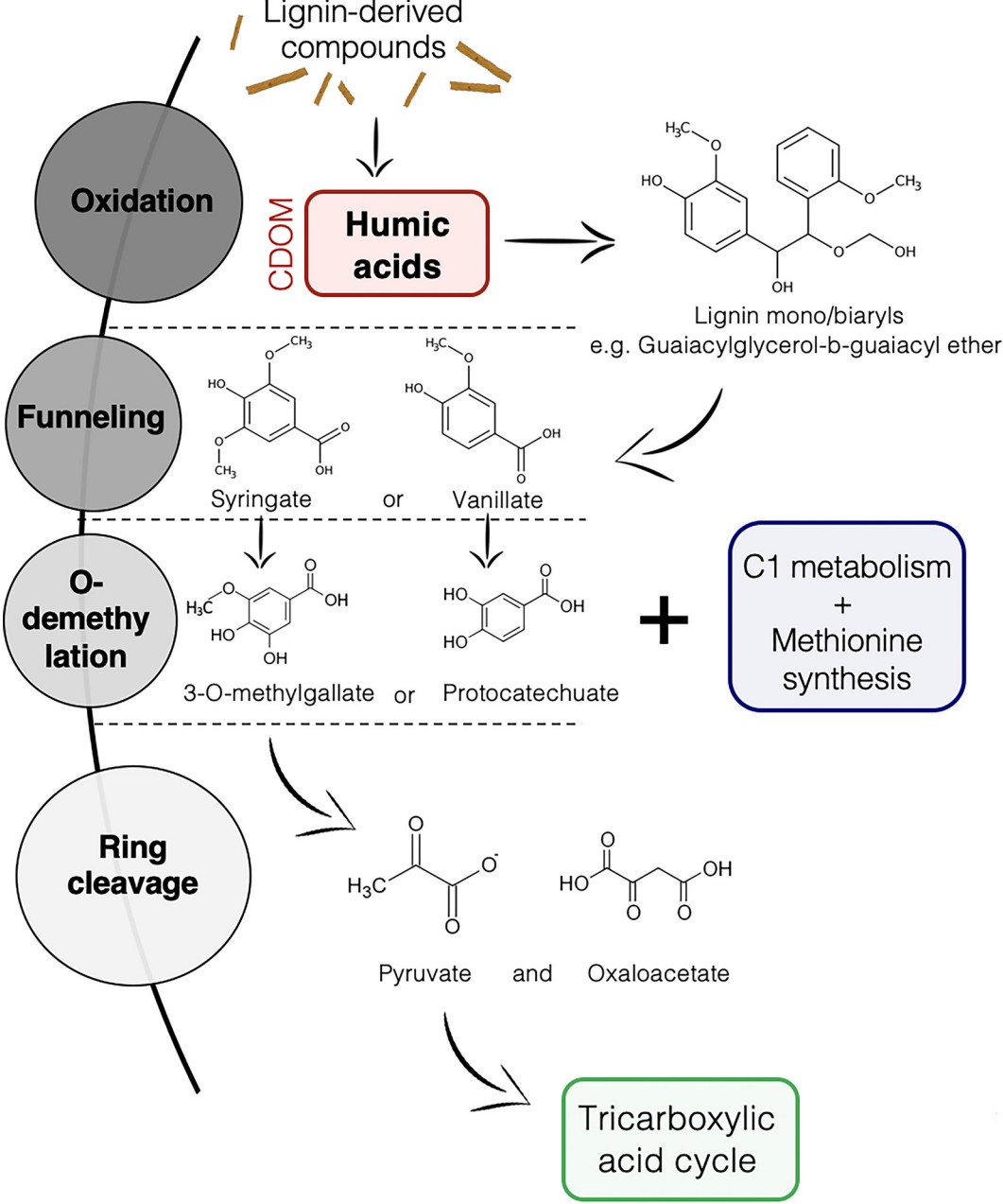

**FIG 1** Main steps of biological degradation of lignin. Humic dissolved organic matter (DOM) (in red) is produced after the initial oxidation of lignin-derived compounds (58, 68). CDOM, chromophoric dissolved organic matter.

the leaching of protons from humic molecules is responsible for acidifying Amazonian black water ecosystems (pH 2.8 to 5), making them physiologically challenging environments that shape the structure of resident aquatic communities (33, 34). Second, we aimed to better understand potential interactions between humic FDOM and the Amazonian bacterioplankton. We hypothesized that the taxa correlated with the abundance of humic FDOM would possess genes potentially involved in its pathways of degradation. To achieve these objectives, we performed a 16S rRNA metabarcoding analysis based on bacterioplankton DNA and RNA extracts to characterize the taxonomic structure of global bacterioplankton (from DNA extracts) and transcriptionally active bacterioplankton (from RNA extracts). In parallel, we assembled 90 Amazonian basin shotgun metagenomes from the literature, to build a tailored functional database used to infer the bacterioplankton functions from the 16S rRNA data.

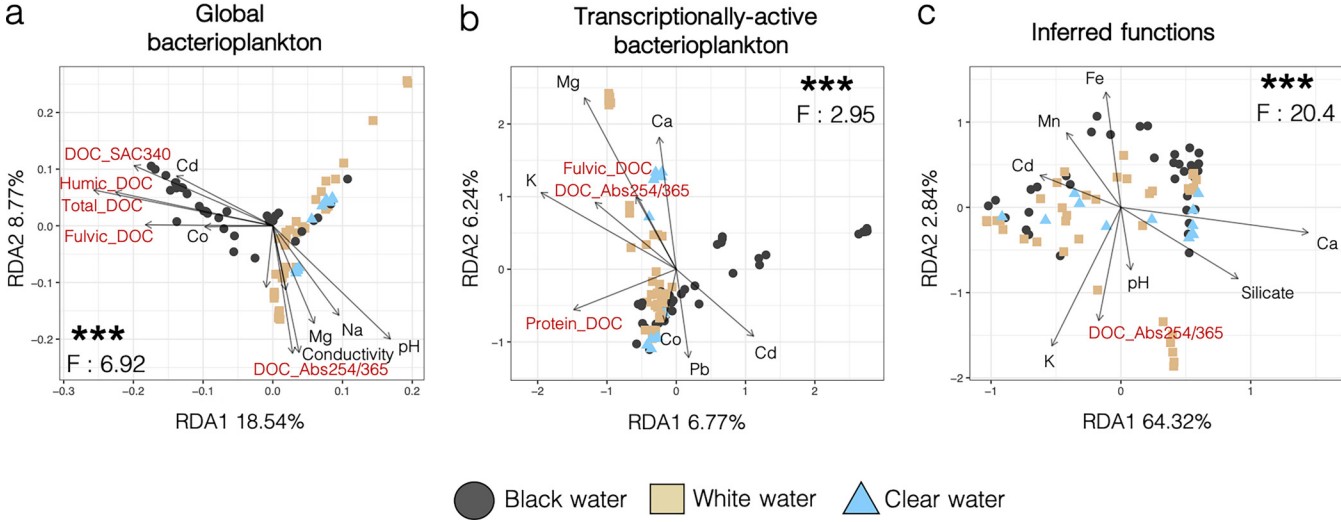

**FIG 2** Distance-based redundancy analyses (RDAs) of global and transcriptionally active bacterioplankton communities, and their functional profiles significantly cluster according to water type. RDA ordination plots of sampling sites according to the global bacterioplankton taxonomic structure (a), transcriptionally active bacterioplankton taxonomic structure (b), and inferred functional repertory (c). The inferred functional repertory is based on the taxonomy of the 16S rRNA amplicon data. The samples are colored according to their water type. The $P$ and $F$ values are from the analysis of variance (ANOVA) tests done on the RDA solution. Environmental parameters associated with dissolved organic carbon (DOC) concentration or fluorescent dissolved organic matter (FDOM) components are in red. ***, $P < 0.001$. Selection of environmental parameters was done according to their variance inflation factors and by using stepwise selection via *ordistep*.

## RESULTS

The Amazonian bacterioplankton showed a rich abundance of *Proteobacteria*, *Actinobacteria*, and *Cyanobacteria* (Fig. S1). While Shannon $\alpha$-diversity did not significantly differ between water types ($P > 0.05$, mean Shannon diversity between 6 and 7 for all water types) (Fig. S2 to S4), $\beta$-diversity analyses showed that bacterioplankton communities significantly clustered according to water type in distance-based redundancy analyses (RDAs) based on the taxonomic structure and inferred functions of these communities (RDA, $P < 0.001$, $F = 2.95$ to 20.4) (Fig. 2). These RDAs (Fig. 2) suggest that white and clear water communities were similar and differed from black water communities. This result was confirmed by the error rates from the confusion matrix of the random forest classification; the rates of misclassification of clear water samples (0.08 to 0.58) were always higher between clear and white than between clear and black water samples (0 to 0.31) (Table S1). The RDAs in Fig. 2a and b show that a higher proportion of the total variance is explained by axes 1 and 2 for the global bacterioplankton (27.31%) than for the transcriptionally active bacterioplankton (13.01%).

The environmental parameters that significantly influenced the taxonomic structure of global bacterioplankton, transcriptionally active bacterioplankton, and the inferred functional repertory of bacterioplankton were not identical. The taxonomic structure of global bacterioplankton from black water was mostly driven by the concentration of $Cd^{2+}$, $Co^{2+}$, humic dissolved organic carbon (DOC), fluvic DOC, total DOC, and levels of SAC340, but in white and clear waters, it was associated with the concentration of $Mg^{2+}$, $Na^+$, and the levels of pH, conductivity, and the absorbance ratio at 254 nm and 365 (Abs254/365). The taxonomic structure of transcriptionally active bacterioplankton in black water was driven by the concentration of $Co^{2+}$, $Pb^{2+}$, and $Cd^{2+}$, but in white and clear waters, the main drivers were the concentrations of $Ca^{2+}$, $Mg^{2+}$, $K^+$, protein-like DOC, fluvic DOC, and the level of Abs254/365. The inferred functional repertory from black water was associated by the concentrations of $Fe^{3+}$, $Mn^{2+}$, and $Cd^{2+}$, while in white and clear waters, it was affected by the concentrations of $Ca^{2+}$ and silicate and the levels of pH and Abs254/365. Overall, the taxonomic structures and inferred functional repertory were driven by several parameters associated with the relative abundance of the different FDOM components (i.e., the relative abundance of humic, fluvic, and protein-like DOC, in addition to the SAC340 and Abs254/365 ratios), which appear in red in the RDAs of Fig. 2.

**TABLE 1** Site identification, water color, geographical coordinates, ecosystem type, and sampling time for each sampling site

| Site no. | Site name | Water color | GPS | | Ecosystem | Sampling time |
| --- | --- | --- | --- | --- | --- | --- |
| | | | South | West | | |
| 1 | Rio Negro – Barcelos | Black | 0°50′50.8′′S | 62°57′40.3′′W | River | November 2018 |
| 2 | Rio Negro – Santo Alberto | Black | 1°23′29.8′′S | 61°59′35.3′W | River | October 2019 |
| 3 | Rio Negro – Anavilhanas | Black | 2°41′46.1′′S | 60°46′33.3′W | River | October 2018 |
| 4 | Lago do Cemeterio | Black | 3°02′16.6′′S | 60°32′42.7′′W | Lake | October 2019 |
| 5 | Lago Téfé | Black | 3°27′55.2′′S | 64°53′13.2′′W | Lake | November 2019 |
| 6 | Rio Branco | White | 1°19′05.7′′S | 61°52′34.7′′W | River | October 2019 |
| 7 | Lago Janauari | White | 3°12′03.4′′S | 60°03′10.1′′W | Lake | October 2018 |
| 8 | Lago Catalão | White | 3°09′56.4′′S | 59°54′38.4′′W | Lake | October 2018 |
| 9 | Lago Janauaca | White | 3°23′37.5′S | 60°19′52.6′′W | Lake | November 2019 |
| 10 | Rio Manacapuru | White | 3°16′16.9′′S | 60°42′03.2′′W | River | November 2018 |
| 11 | Lago Téfé-Solimões | White | 3°21′07.4′′S | 64°40′21.4′′W | Lake | November 2019 |
| 12 | Lago des Pirates | White | 3°15′19.2′′S | 64°41′44.3′W | Lake | November 2019 |
| 13 | Balbina Reservoir | Clear | 1°50′55.9′′S | 59°34′59.5′W | Reservoir | October 2018 |
| 14 | Rio Tapajós | Clear | 2°18′57.8′′S | 55°00′45.0′′W | River | October 2019 |
| 15 | Rio Curua-Una | Clear | 2°48′19.1′′S | 54°17′52.2′′W | River | November 2018 |

Permutational multivariate analysis of variance (PERMANOVA) analyses (999 permutations) have shown that water type was significantly associated with the taxonomic structure of global bacterioplankton ($F = 8.5$, df.res = 82, $R^2 = 0.17$, $P < 0.001$), transcriptionally active bacterioplankton ($F = 2.9$, df.res = 82, $R^2 = 0.06$, $P < 0.001$), and the inferred functional repertory ($F = 11.8$, df.res = 82, $R^2 = 0.22$, $P < 0.001$). Analyses of variance (ANOVAs) performed on overall RDA solutions were also consistently significant (Fig. 2). Betadisper permutests (1,000 permutations) showed homogenous variance between groups for the inferred functions data set ($F = 0.73$, df.res = 167, $P = 0.45$). They have shown heterogenous variance for the taxonomic structure of global ($F = 13.23$, df.res = 82, $P < 0.001$) and transcriptionally active bacterioplankton ($F = 4.79$, df.res = 82, $P = 0.01$); however, they also showed in both cases that the largest variance occurred in the group with the largest number of samples (i.e., white water). In this situation, PERMANOVA tests are known to be overly conservative, especially with unbalanced designs (35), but still showed a significant signal between water types in our case (Fig. 2; Fig. S3). In addition, PERMANOVA tests suggested that communities' composition from water bodies characterized by different water residence time (lake versus river systems) significantly differed (all $P$ values $< 0.02$) according to the Ecosystem variable from Table 1 (Table S2 and the supplemental results and discussion in the supplemental material).

We implemented a machine-learning random forest algorithm to identify which taxa best discriminated different water colors. The out-of-bag error rate (node error rates in the trees of classification) was only 1.18% when considering the taxonomic structure of global bacterioplankton, 27.06% for transcriptionally active bacterioplankton, and 23.53% for inferred functions (Table S1). For global bacterioplankton, the amplicon sequence variants (ASVs) that best discriminated water colors were from the *Proteobacteria*, *Alphaproteobacteria*, *Actinobacteria*, *Gammaproteobacteria*, and *Polynucleobacter* clades (ASVs were annotated to the best taxonomic resolution possible). Black water samples were mostly characterized by an enrichment of *Gammaproteobacteria*, *Actinobacteria*, and *Polynucleobacter* (Fig. 3). For transcriptionally active bacterioplankton, discriminant ASVs were taxonomically more diverse and represented members of *Bradyrhizobium*, *Actinobacteria*, *Ralstonia*, *Delftia*, *Geobacillus*, *Commamonadaceae*, *Alphaproteobacteria*, *Burkholderiales*, *Caulobacteraceae*, and *Polynucleobacter*. Black water samples were mostly characterized by an increased abundance of *Alphaproteobacteria*, *Burkholdariales*, *Caulobacteraceae*, *Actinobacteria*, and *Polynucleobacter*. ASVs from the clade *Polynucleobacter* possessed the highest taxonomic resolution and were consistently associated with black water samples in global and transcriptionally active communities (Fig. 3).

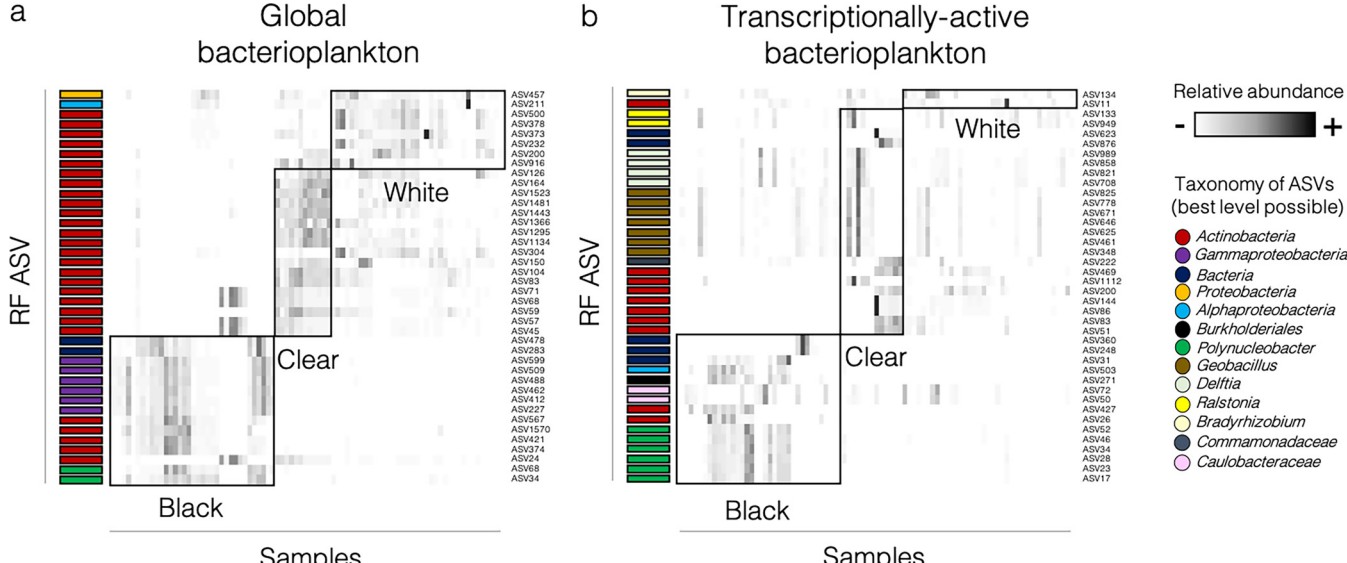

**FIG 3** Random forest (RF) machine-learning analyses identified the 40 amplicon sequence variants (ASVs) showing the most important differentiation between water types. The results are shown in heat maps for the taxonomical structure of the global bacterioplankton community (a) and the transcriptionally active bacterioplankton community data set (b). The heat map columns represent samples, and rows are different ASVs. The taxonomic annotation of each ASV was done at the highest resolution possible for each ASV. The colored tag at the left of each row corresponds to the ASV taxonomic assignation. Black boxes identify to which water type the taxa were associated (i.e., in which water type they were most abundant). The relative abundance (color intensity) of each ASV in each sample is scaled according to the abundance of the same ASV in all other samples.

The relative abundance of the different FDOM components was specific to each water type of the Amazon basin (Fig. 4a; Table 2; Fig. S5). The parallel factor analysis (PARAFAC) model showed that the FDOM was composed of three main fractions: the humic-like, fulvic-like, and protein-like fractions (Fig. 4b to d). Black water sites contained higher concentrations of DOC, with FDOM profiles significantly enriched ($P = 0.006$, $t = 3.24$, df = 13) in the humic-like fraction of greater aromaticity and molecular weight. White water sites contained FDOM characterized by a high content of fulvic-like components, while clear water sites contained more protein-like FDOM of low aromaticity and molecular weight (see Table 2 for raw data; Fig. 4; FEEM scans on Fig. S5).

The humic-like fraction of FDOM was associated with the taxonomic structure of global and transcriptionally active bacterioplankton communities within the different waters of the Amazon basin, as the humic FDOM relative abundance isolines respected the natural clustering of the communities on the redundancy analyses of Fig. 5a and b. A coabundance Spearman correlation network-based approach enabled us to identify which taxonomic groups were significantly (Bonferroni-corrected $P < 0.05$) correlated with humic FDOM (Fig. 5c and d): *Acetobacteraceae*, *Polynucleobacter*, *Methylocystis*, and several unidentified Beta- and Gammaproteobacteria. The results for global and transcriptionally active bacterioplankton showed different correlation profiles. Indeed, for global bacterioplankton, the results suggested a direct correlation between humic FDOM concentration and 44 taxa. In contrast, for transcriptionally active communities, the influence seemed to be mostly indirect, since there were only three taxa directly correlated with humic FDOM relative abundance. However, these three taxa (two *Polynucleobacter* and one *Acetobacteraceae* ASVs) were key in the overall community transcriptionally active community; they were important interaction hubs, as their activity was strongly correlated with 93 other taxa.

We recomputed the Spearman correlation analysis (between humic FDOM and bacterial ASV) after agglomerating all ASVs at the genus level to ensure compatibility with the metagenome reference database. Several taxa that were significantly associated with humic FDOM at the ASV level were also associated with humic FDOM at the genus level (e.g., *Polynucleobacter* and *Methylocystis*). We investigated the presence/absence of the enzymes known to be part of the humic FDOM degradation pathways (Table S3)

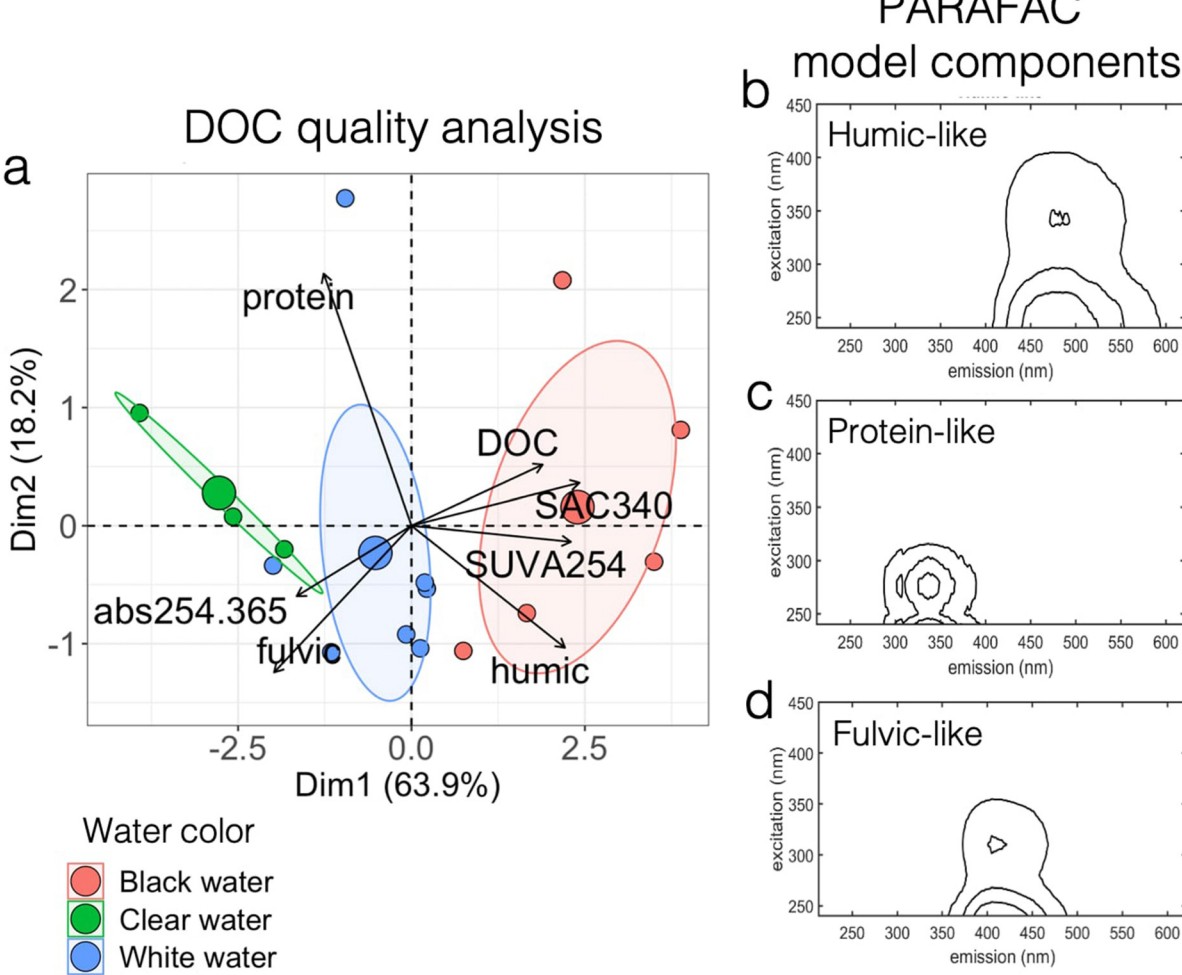

**FIG 4** Black water sites showed unique DOC and FDOM profiles, characterized by a higher relative abundance of aromatic, high molecular weight humic carbon. (a) Principal component analysis showing how sampling sites cluster according to their FDOM optical characteristics. The three bigger dots refer to group centroids. The environmental variables are as follows: "DOC" refers to concentration (mg/liter) of DOC; "protein" refers to percentage of protein-like FDOM; "humic" refers to percentage of humic-like FDOM; "fulvic" refers to percentage of fulvic-like FDOM; and SAC340, SUVA254, and abs254.365 refer to absorbance ratios detailed in Table 2. (b to d) Parallel factor analysis (PARAFAC) model components show the presence of humic-like (b), protein-like (c), and fulvic-like (d) FDOM fractions in the sites sampled.

in the subset of all genera in which abundance correlated with humic FDOM (Fig. 6) and found that three genera were significantly correlated with humic FDOM degradation pathways: *Polynucleobacter*, *Methylobacterium*, and *Acinetobacter*. When detected, significant Spearman correlations varied between 0.30 and 0.51. These three genera possessed enzymes involved in all four main steps of the degradation of humic compounds.

Four main results suggest that among all taxa, the genus *Polynucleobacter*, which had a relatively low abundance (mean of 0.05% to 2.25%) in global and transcriptionally active bacterioplankton (Fig. S6), showed the strongest association to humic FDOM. First, the random forest analysis has shown that *Polynucleobacter* was one of the best taxa to discriminate the taxonomic structure of black water samples (rich in humic FDOM) from white/clear water samples (poor in humic FDOM), in both global and transcriptionally active bacterioplankton (Fig. 3). Second, *Polynucleobacter* ASVs abundances were significantly correlated with humic FDOM concentrations (Bonferroni-corrected $P$ value < 0.05) in global and transcriptionally active bacterioplankton communities (in which they represented two of three ASVs correlated with humic FDOM) (Fig. 5c and d). Third, when correlation analyses were performed at the genus level, *Polynucleobacter* was the genus of

**TABLE 2** Measure of DOC quantity and FDOM optical characteristics[a]

| Site no. | Water color | DOC concn | SAC340 | SUVA254 | Abs254/365 | Humic FDOM (%) | Fulvic FDOM (%) | Protein FDOM (%) |
|---|---|---|---|---|---|---|---|---|
| 1 | Black | 10.9 | 39.5 | 4.5 | 3.8 | 56.7 | 29.5 | 13.8 |
| 2 | Black | 11.7 | 33.5 | 3.7 | 3.6 | 60.3 | 32.6 | 7.1 |
| 3 | Black | 11.4 | 30.5 | 3.6 | 3.8 | 47.2 | 30.3 | 22.5 |
| 4 | Black | 9.8 | 18.9 | 2.4 | 3.8 | 52.3 | 41.0 | 6.7 |
| 5 | Black | 7.1 | 29.1 | 3.4 | 4.0 | 54.2 | 37.3 | 8.5 |
| 6 | White | 6.0 | 19.1 | 2.2 | 4.3 | 50.8 | 39.7 | 9.5 |
| 7 | White | 7.1 | 19.1 | 1.4 | 2.2 | 34.7 | 36.2 | 29.1 |
| 8 | White | 9.1 | 11.7 | 2.1 | 6.4 | 37.5 | 45.6 | 16.8 |
| 9 | White | 5.7 | 20.0 | 2.6 | 4.2 | 50.6 | 40.4 | 9.0 |
| 10 | White | 8.0 | 22.1 | 3.0 | 4.6 | 46.2 | 41.8 | 12.0 |
| 11 | White | 5.7 | 20.1 | 2.6 | 3.8 | 49.0 | 39.3 | 11.7 |
| 12 | White | 6.5 | 14.2 | 2.2 | 4.7 | 43.9 | 45.3 | 10.8 |
| 13 | Clear | 4.9 | 6.1 | 1.2 | 7.1 | 30.6 | 42.3 | 27.1 |
| 14 | Clear | 2.7 | 8.7 | 1.9 | 5.0 | 44.8 | 38.3 | 16.9 |
| 15 | Clear | 4.6 | 11.7 | 1.9 | 5.3 | 35.1 | 45.0 | 20.0 |

[a]SAC340 and SUVA254 are the specific absorbance coefficients index of relative DOM aromaticity (the higher the values, the more aromatic is the DOM). Abs254/365 is the index of molecular weight (the lower the value, the higher the molecular weight of the DOM). DOC, dissolved organic carbon; DOC concn, DOC concentration in mg/liter; DOM, dissolved organic matter; FDOM, fluorescent dissolved organic matter.

which the relative abundance showed the strongest correlation (Spearman correlation = 0.30 to 0.51) to the presence of humic FDOM degradation pathways (Fig. 6). Fourth, the inferred functional repertory of Amazonian *Polynucleobacter* suggested that this group had the genetic potential to be involved in all four main degradation steps of humic compounds (Fig. 6): initial oxidation, funneling, *O*-demethylation, and ring cleavage pathways.

## DISCUSSION

**Water type shapes the structure of bacterial communities and their inferred functional potential.** Water type has been shown to be a major driver of the diversity, composition, and population genomics of eukaryotic biological communities in Amazonia. This has been shown in a vast array of species, including teleosts (36–39), phyto and zooplankton (40), and periphyton communities (41). Our results showed that the Amazonian bacterioplankton was similar in composition to what has been previously reported (22), with an important influence of water type on the taxonomic structure of global and transcriptionally active bacterioplankton communities and on their functional profiles (Fig. 2). At the taxonomic and functional levels, we showed that among the environmental parameters that were the most associated with community clustering (DOC quantity and type, pH, conductivity, and concentrations of $Cd^{2+}$, $Co^{2+}$, $Mg^{2+}$, $Na^+$, $Ca^{2+}$, $K^+$, $Pb^{2+}$, $Fe^{3+}$, $Mn^{2+}$, and silicates), several are known to be the main parameters driving differences between water types. For instance, higher concentrations of total DOC and overall relative enrichment in humic FDOM are known to be strongly associated with black waters (18, 42). Interestingly, we observed that the clustering of functional repertory according to water types (Fig. 2c) was not as clear as structural or transcriptional activity profile clustering (Fig. 2a and b). This result might be associated with the fact that several housekeeping genes are shared by all bacterial members, thus reducing intersite variability when considering the functional repertories. However, based on the significant PERMANOVA results, it still appears that there were significant water type-specific functional profiles.

**Humic FDOM and bacterioplankton communities.** Dissolved organic carbon forms the very basis of the majority of aquatic food webs and is an important food source to heterotrophs within river systems (43, 44). The bioavailability of DOC to bacterioplankton depends on the type of DOC present. It has been suggested that allochthonous humic-like DOC may be more bioavailable to bacteria than lower molecular weight DOC (45) (although research on saltwater ecosystems has shown different results [46]) and that some bacteria prefer terrestrially derived DOC over autochthonous protein-like DOC derived from algae and/or bacteria (47). Previous research has shown that bacteria are able to breakdown humic DOC, supporting the idea that this

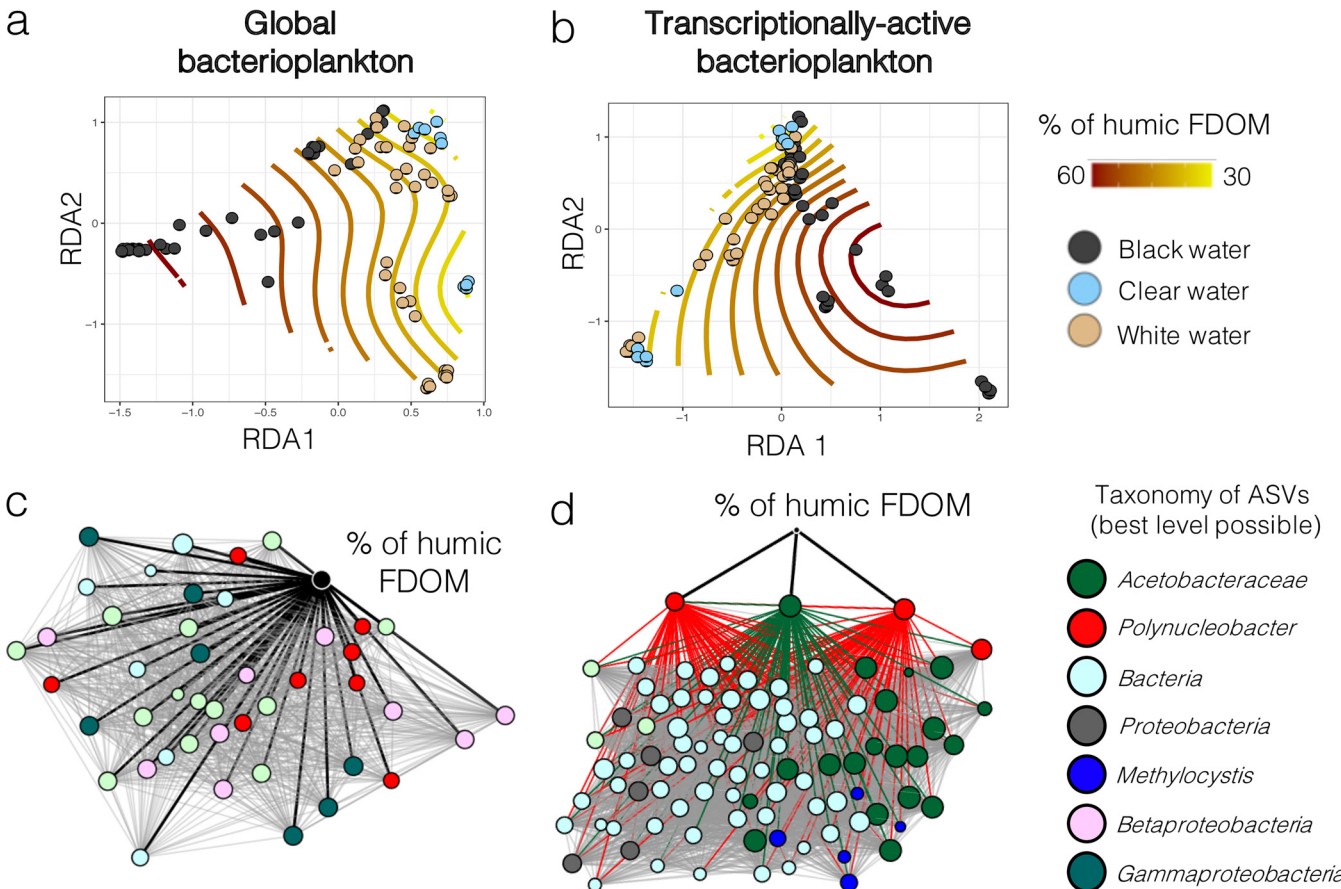

**FIG 5** The concentration of humic FDOM correlated with the structure of global and transcriptionally active bacterioplankton communities. (a, b) Ordisurf plots (i.e., RDA plots fitted with isolines describing humic FDOM relative abundance variations) based on the structure of global (a) and transcriptionally active bacterioplankton (b). The samples are colored according to their water type. (c, d) Spearman correlation-based networks of coabundance between humic FDOM and bacterial ASVs from the global bacterioplankton (c) or transcriptionally active bacterioplankton (d) communities. For ease of viewing, only the taxa that correlated directly with humic FDOM or that were direct neighbors of such taxa were kept for network construction. The nodes are colored according to their taxonomic assignation (at the best level possible). Labels in the legend for panels c and d refer to the best level of taxonomic assignation reached for these ASVs. The black lines in panels c and d represent direct correlations between humic FDOM and specific ASVs, while gray lines represent indirect correlations. All correlations in panels c and d are positive correlations. In panel d, lines from the three nodes directly correlated with humic FDOM are colored according to their node taxonomic assignation to highlight the indirect effect of humic FDOM on the transcriptionally active bacterioplankton community. These lines sre not colored in panel c for ease of viewing. FDOM, fluorescent dissolved organic matter; ASV, amplicon sequence variant.

component is bioavailable to some bacterial species (45, 47). Our analysis of the FDOM fractions from the 15 sites showed that black waters have higher concentrations of DOC but also show distinct FDOM profiles comprising a significant enrichment in the humic fraction characterized by higher SAC340 and SUVA254 scores (Table 2; Fig. 4). These results support previous findings (18, 42) suggesting that naturally acidic waters show a unique FDOM signature compared to circumneutral and groundwater-fed systems. Multivariate correlation analyses also suggested that the relative abundance of humic FDOM is an important factor shaping the taxonomic structure of global and transcriptionally active bacterioplankton (Fig. 2 and 5). Furthermore, analyses at the functional level showed that there are several genera in the Amazonian bacterioplankton community of which the abundance correlates with humic FDOM and that possess genes associated with its degradation processes (Fig. 6).

Overall, in our data set, the genus *Polynucleobacter* has shown the strongest correlation to the relative abundance of humic FDOM (Fig. 6). The genus *Polynucleobacter* mostly comprises free-living aquatic bacteria and is omnipresent in freshwater lakes and ponds worldwide (48), including in several Amazonian streams (23, 24). Several studies have detected a strong correlation between the abundance of this genus and DOC concentrations (48–50). Furthermore, another study has shown that *Polynucleobacter* subclusters

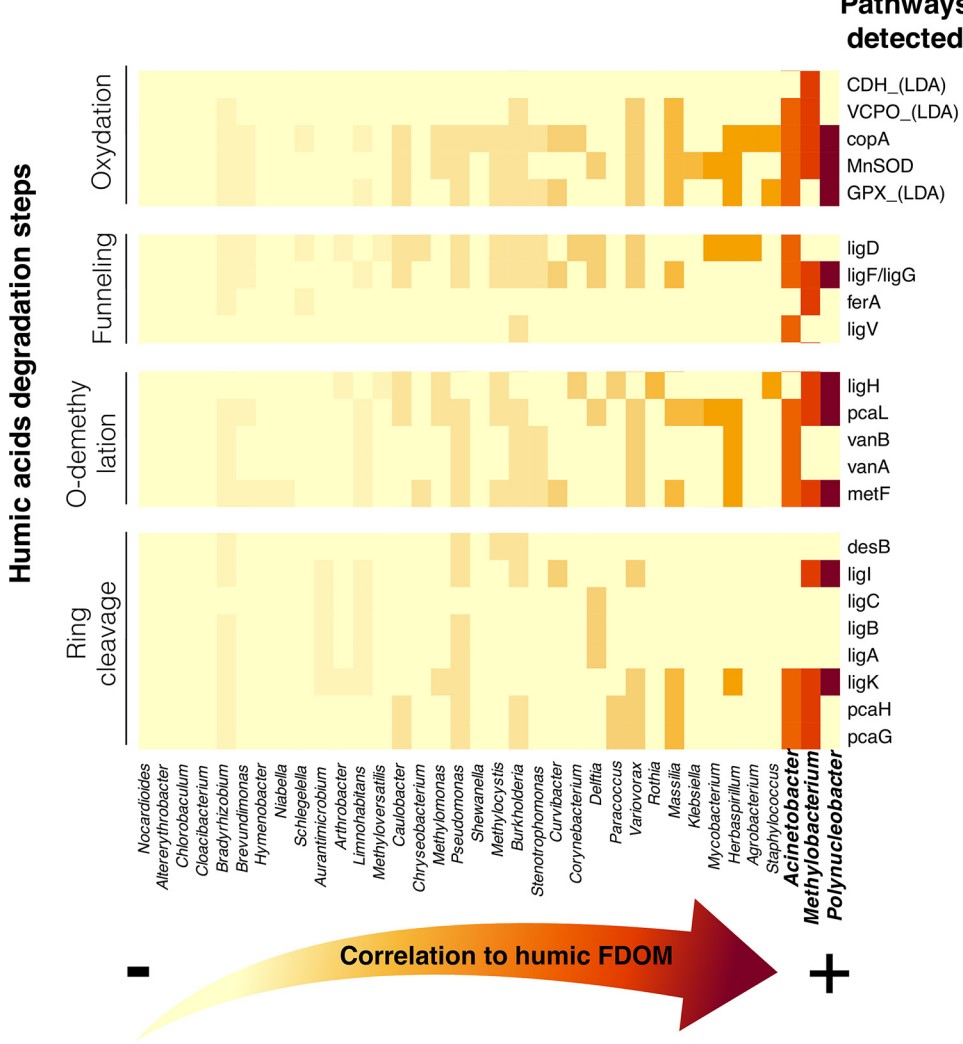

**FIG 6** Potential presence of different pathways of the main steps of humic acids' degradation in the genera that abundance correlated with humic FDOM. Spearman correlation between the relative abundances of humic FDOM and bacterial genera are plotted on a heat map from least to most correlated (left to right, respectively). *Polynucleobacter* is the genus showing the highest correlation to humic FDOM concentration, followed by *Methylobacterium* and *Acinetobacter*. The complete list of pathways that were screened for in the functional repertory can be found in Table S3; however, only the pathways that were found in at least one genus are represented in the heat map for ease of viewing. FDOM, fluorescent dissolved organic matter.

show ecological niche separation in accordance with DOM optical characteristics (51). *Polynucleobacter* phylotypes respond quickly to an enhanced availability of DOM (52), and some reports suggest that *Polynucleobacter* can feed on it (49, 50). However, experiments conducted on a population of *Polynucleobacter* from a humic temperate pond indicated that these bacteria mostly live as chemoorganotrophs by utilizing low-molecular-weight substrates derived from the photooxidation of humic substances (53), as suggested in other studies (50, 54). The high growth of *Polynucleobacter* phylotypes (55) could be favored by the chemical mineralization of low-molecular-weight substances such as acetate, a typical photolysis product (54). Overall, the genomic potential to degrade humic compounds or humic by-products appears to be strain specific in *Polynucleobacter*. Indeed, Hahn and colleagues (53) did not detect genes involved in humic substances degradation (i.e., mono- and dioxygenases) but found a pathway encoding the degradation of humic compounds in another strain of *Polynucleobacter asymbioticus* (56).

In our study, it is unlikely that the ASVs identified as *Polynucleobacter* originated from only one species; for instance, a recent study detected an important diversity of

this genus: 60 to 90 species-like *Polynucleobacter* operational taxonomic units (OTUs) were detected in temperate rivers (57). Until now, the implication of members of this clade in the degradation of humic compounds in Amazonia has yet to be investigated. Although we did not measure specific humic degradation rates in this study, the set of genes that was detected in Amazonian *Polynucleobacter* (Fig. 6) suggests that they possess the genomic potential to be involved in the degradation of humic substances such as humic acids or their by-products via a derivative of the $\beta$-aryl ether degradation pathway for diaryl residues (58).

In addition to *Polynucleobacter*, our results suggest that *Methylobacterium* and *Acinetobacter* are also strongly correlated with the relative abundance of humic FDOM. They also show that these clades possess genes coding for enzymes involved in all the main steps of humic degradation (Fig. 6). Like *Polynucleobacter*, these genera possess the genomic potential to degrade humic substances via a derivative of the $\beta$-aryl ether degradation pathway for diaryl residues. *Methylobacterium* could also profit from by-products of the humic degradation; methanol, one of the main carbon sources for methylotrophic bacteria like *Methylobacterium* (59), is produced during the demethylation of humic substances (60). Several studies have documented the humic substances degradation potential of *Acinetobacter* (61, 62). The set of genes detected in this genus suggests that the humic compounds' *O*-demethylation (63) and ring cleavage (64) differ from those of *Polynucleobacter* and *Methylobacterium* (see more details in the supplemental material under "Pathways of humic compounds' degradation"). Interestingly, while *Polynucleobacter* is exclusively aquatic, *Methylobacterium* and *Acinetobacter* are often found in humic soils (65, 66) and could potentially be derived from terrestrial organic matter that leached into the riverine water.

Since not all enzymes required for the complete degradation of humic compounds were detected in the aforementioned genera, further studies are needed to decipher whether and exactly how these bacteria degrade humic DOC and to determine whether they are able to perform this process alone, with associations with other members of the bacterioplankton community, or with nonbacterial microbes. Although not tested here, functional compartmentalization of the community for humic degradation has already been documented (67) and could potentially involve interkingdom relationships via alternative pathways that remain to be discovered (reviewed in references 58 and 68). Finally, further studies should also assess whether bacterial DOC degradation in Amazonia is dependent on a coupling with physical photodegradation processes, as suggested by Nalven et al. (69).

**Conclusions.** The results from our study show that the most important environmental factors affecting the Amazonian bacterioplankton communities within the three different water types of the Amazon basin rely on the relative abundance of the different FDOM fractions detected, especially the enrichment in humic FDOM characteristic of black water environments. Among the taxa mostly associated with the different water types, ASVs assigned to the genera *Polynucleobacter*, *Methylobacterium*, and *Acinetobacter* particularly stood out, as their relative abundance in global and transcriptionally active bacterioplankton was strongly associated with black water environments and correlated well with the relative abundance of humic FDOM. The inferred functions of these genera suggest that they possess genes coding for enzymes implicated in the main degradation steps of humic compounds, indicating that the role of these taxa in carbon cycling within the Amazonian basin merits further investigation.

## MATERIALS AND METHODS

**Sampling and processing.** Water samples were collected from 15 sites in the Brazilian Amazon basin in October and November of 2018 and 2019 (sampling times indicated in Table 1). The 15 sites were distributed over an area of >300,000 km² along the Rio Negro, Rio Solimões, and Rio Tapajos watersheds, the three major tributaries of the Amazon River (10). GPS coordinates and a map of all sites are found in Table 1 and Fig. 7, respectively. The 15 sites include five black water, seven white water, and three clear water sites. Six replicate water samples were collected per site. Surface water samples were taken at a depth of 30 cm in 2-liter Nalgene (Thermo Fisher Scientific, Waltham, MA) bottles. Filtration was performed as previously described (70) through 0.22-$\mu$m-pore size polyethersulfone Sterivex filters

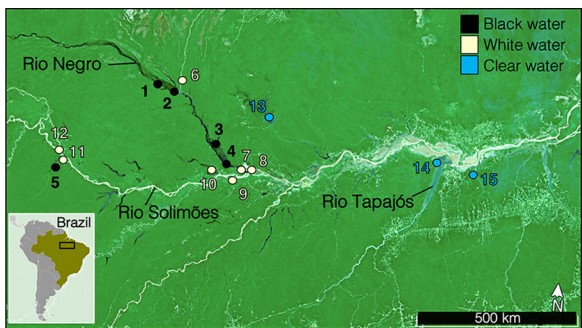

**FIG 7** Map of the 15 sampling sites distributed throughout the Brazilian Amazon basin. The colors of the dots represent the water type (black, white, or clear).

(catalog no. SVGP01050, Millipore, Burlington, MA) less than 30 min after collection. Immediately upon collection, the filters were stored in 2 mL of nucleic acid preservation (NAP) buffer. NAP buffer, which preserves DNA and RNA integrity at room temperature, contains EDTA disodium salt dihydrate, sodium citrate trisodium salt dihydrate, and ammonium sulfate (71, 72). The samples were then stored in $-80°C$ until processing. Before DNA/RNA extraction, the Sterivex filter casings were opened and processed according as previously described (70) using sterile instruments, and the filter membranes were stored in TRIzol (catalog no. 15596026, Thermo Fisher Scientific). DNA and RNA extractions were performed according to the manufacturer's instructions for TRIzol without modification. Four blank controls (sterile filters stored in the NAP buffer) were also processed identically to all samples for DNA/RNA extractions.

**Environmental variables.** A total of 34 environmental variables commonly characterized in limnological studies (73) and associated with the physicochemical differences between different water colors (10) were measured in this study (Table 2; Table S4 to S6). Temperature (°C), conductivity ($\mu$S), pH, and dissolved oxygen (%) were measured directly on-site using a YSI professional plus series multimeter (YSI Inc./Xylem Inc., Yellow Springs, OH). The concentration of DOC, dissolved metals, nutrients, free ions, and chlorophyll $a$ were measured at the laboratory, according to the techniques described below.

**Chlorophyll $a$ and phaeopigments.** Three replicates of 250 mL of water per site were filtered using a Masterflex Easy-Load II peristaltic pump from Cole-Parmer (catalog no. HV-77200-62, Montreal, Quebec, Canada) through 0.45-$\mu$m-pore size glass fiber filters, which were then immediately stored at $-80°C$ for chlorophyll $a$ quantification (74). Chlorophyll $a$ was extracted after a 24 h of incubation of filters in acetone at $-20°C$ before measuring the absorbance on a Turner Designs (San Jose, CA) fluorimeter model 10 AU (catalog no. 1100-100). Chlorophyll and phaeopigment concentrations were then calculated according to the method described previously (74, 75).

**Nutrients.** For nutrients ($NO_2^-$, $NO_3^-$, and silicates) analysis, three replicates/site of 12 mL of water were filtered through 0.45-$\mu$m-pore size glass fiber filters and stored in sterile and acid-washed (1 M HCl) 15-mL Falcon tubes (Fisher Scientific, Hampton, NH). A total of 24 $\mu$L of $HgCl_2$ solution (1 g/100 mL) was added for conservation, before measurement on a Bran and Luebbe III (AA3) nutrient autoanalyzer as previously described (76).

**(i) Ionic composition.** Water samples for determination of ionic composition ($Na^+$, $Ca^{2+}$, $Mg^{2+}$, and $K^+$) were analyzed using flame atomic absorption spectroscopy (PerkinElmer model 3100, catalog no. 63929-1, PerkinElmer Inc., Woodbridge, Ontario, Canada). $Cl^-$ was measured using the colorimetric method described previously (77). Hardness was calculated from the $Ca^{2+}$ and $Mg^{2+}$ concentrations.

**(ii) Dissolved organic carbon and dissolved metals.** For analysis of dissolved metals and DOC, we filtered samples through Millipore polyvinylidene difluoride (PVDF) 0.45-mm Sartorius filters (Sartorius, Germany). Metal suites were measured by inductively coupled plasma mass spectroscopy (ICP-MS). Quality assurance and quality control (QA/QC) consisted of analysis of method blanks, laboratory duplicates, and matrix spikes conducted using certified standards. Total suspended solids was determined using gravimetric analysis as outlined in Environmental Protection Agency Method 160.2 (78). DOC was analyzed as nonpurgeable organic carbon using a total carbon analyzer (Apollo 9000 combustion TOC analyzer: Teledyne Tekmar, Mason, OH). The TOC machine was calibrated using primary standard grade potassium hydrogen phthalate (KHP), and QA/QC KHP standards were run every 10 samples. Fluorescence excitation emission (FEEM) and absorbance scans were performed using a quartz cuvette in an Aqualog fluorimeter (HORIBA Scientific, Piscataway, NJ) to determine FDOM components and characteristics. FEEM scans along with simultaneous absorbance measurements were conducted on all samples, with excitation wavelengths in 2-nm steps between 250 and 450 nm, and emission wavelengths of 250 to 620 nm. The absorbance of a blank of ultrapure water was run before each sample and automatically subtracted from each sample, with inner filter effects and first and second order Rayleigh and Raman scatter also removed. The FEEMs were analyzed in MATLAB R2014b (MathWorks, Inc.) and modeled using PARAFAC (PLS-toolbox in MATLAB; Eigenvectors Research Inc., Manson, WA). The PARAFAC model was validated following the previously described recommendations (79). No clear consistent patterns and peaks were visible in the residual plots, core consistency was 99%, and split-half analysis results were consistent with the model. Relative DOC aromaticity and molecular weight were determined using specific UV absorbance at 254 nm (SUVA254) and at 350 nm (SAC340) along with Abs254/365. The raw data of the excitation emission and the absorbance scans are available on OSF (https://osf.io/dz6vf).

The water type (Table 1) of each site was determined based on the physicochemical profiles of the sampled environments, which mostly differed in terms of pH, DOC quantity, FDOM optical characteristics, and conductivity. Black waters are usually acidic due to a high load of humic DOC and also have a very low conductivity (15). White waters have a circumneutral pH, and they usually have lower overall DOC concentrations (enriched in fulvic-like DOC) but high conductivity levels. Clear waters have a slightly acidic pH, with low DOC concentrations (enriched in protein-like DOC) and low conductivity (10). The water types are also easily recognized by their appearance: black waters are rich in tannins that stain the water like tea, white waters have a milky appearance due to the very high load of suspended sediments, while clear waters are transparent. Our measurements of environmental parameters confirmed the *a priori* knowledge of the water types of these sampling sites, based on the literature.

**16S rRNA sequence analysis.** The taxonomic structure of global and of transcriptionally active bacterial communities were assessed using 16S rRNA approaches, respectively, conducted on DNA and RNA extracts. Retrotranscription of the RNA extractions was done using the qScript cDNA synthesis kit (catalog no. 95048-100) from QuantaBio (Beverly, MA) according to the manufacturer's instructions. Then, the fragment V3-V4 (~500 bp) of the 16S rRNA gene was amplified from DNA and cDNA extracts by two PCRs. The first PCR was performed with primers specific to the V3-V4 region of the 16S rRNA gene (primers 347F and 803R) (80), which were tailed on the 5′ end with part of the Illumina TruSeq adaptors (oligonucleotide sequences; Illumina, Inc.). The following oligonucleotide sequences were used for amplification for the first PCR (actual primer sequences are in bold, and the rest corresponds to the adapter sequence): forward (347F), 5′-ACACTCTTTCCCTACACGACGCTCTTCCGATCT**GGAGGCAGCAGTRRGGAAT**-3′; and reverse primer (803R), 5′-GTGACTGGAGTTCAGACGTGTGCTCTTCCGATCT**CTACCRGGGTATCTAATCC**-3′. Then, a second PCR was performed to attach remaining adaptor sequence (the regions that anneal to the flowcell and library specific barcodes): generic forward second PCR primer, 5′-AATGATACGGCGACCACCGAGATCTACAC[index1] ACACTCTTTCCCTACACGAC-3′; and generic reverse second PCR primer, 5′-CAAGCAGAAGACGGCATACGAGAT [index2]GTGACTGGAGTTCAGACGTGT-3′.

The PCRs were performed according to the manufacturer's instructions of the Qiagen Multiplex PCR kit (catalog no. 206143, Hilden, Germany) using an annealing temperature of 60°C and 30 amplification cycles. Amplified DNA was purified with AMPure beads (catalog no. A63880; Beckman Coulter, Pasadena, CA), according to the manufacturer's instructions, to eliminate primers, dimers, proteins, and phenols. Post-PCR DNA concentration and quality were assessed on a Qubit instrument (Thermo Fisher Scientific) and by electrophoresis on 2% agarose gels. After purification, multiplex paired-end sequencing was performed on Illumina MiSeq by the Plateforme d'Analyses Génomiques at the Institut de Biologie Intégrative et des Systèmes of Université Laval.

After sequencing, 24,341,734 sequences were obtained (mean of 135,232 sequences/sample). DADA2 (81) was used for ASV picking. Quality control of reads was done with the *filterAndTrim* function using the following parameters: 290 for the forward read truncation length, 270 for the reverse read truncation length, 2 as the phred score threshold for total read removal, and a maximum expected error of 2 for forward reads and 3 for reverse reads. The filtered reads were then fed to the error rate learning, dereplication, and ASV inference steps (with default settings) using the functions *learnErrors*, *derepFastq*, and *DADA*, which are all from the DADA2 pipeline (81). The merging of sequence pairs was done using *mergePairs* (with default settings) also from DADA2. Chimeric sequences were removed using the *removeBimeraDenovo* function (default settings) with the "consensus" method parameter. Sequenced PCR negative controls were used to remove ASVs identified as potential cross contaminants using the *isContaminant* function from the "decontam" package in R with a default threshold of 0.4. Taxonomic annotation of amplicon sequence variants (ASVs) was performed by using *blastn* matches against NCBI "16S Refseq Nucleotide" database (November 2020). As the NCBI database for 16S sequences is updated more frequently than other sources (Silva, Greengenes, etc.), it matched our requirements for exhaustive information about lesser-known taxa while minimizing ambiguous annotations. Matches above 99% identity were assigned the reported taxonomic identity. Sequences with no matches above the identity threshold were assigned taxonomy using a lowest common ancestor method generated on the top 50 matches using *blastn* (E value cutoff of 1e−4 with default parameters). This method is closely inspired from the Lowest Common Ancestor (LCA) algorithm implemented in MEGAN (82). An analysis of Shannon diversity according to sampling depth for each sample can be found in Fig. S2 and S4. ASV tables, metadata files, and taxonomy information were incorporated into *phyloseq* objects (83) (phyloseq v. 1.32.0) before downstream analyses.

**Shotgun metagenome functional database.** We built a reference metagenomic database to infer the functional profile of the microbial communities previously characterized using the 16S rRNA approach. The reference database was built from 90 high quality metagenomes from the Amazon River sampled in previous studies (21, 22, 24, 27, 29) that included samples from the upper and lower parts of the basin, as well as the Amazon River plume. These shotgun metagenomes were previously compiled in the Amazon River basin Microbial nonredundant Gene Catalogue (AMnrGC) by Santos et al. (21). Details on the environments where these metagenomes were collected are provided as supplemental data by Santos et al. (21). The shotgun metagenomes (see accession numbers under Data availability) were fetched from the NCBI Sequence reads archive (SRA). *Trimmomatic* unpaired filtered reads (default parameters) were assembled using the megahit assembler in a coassembly with the large metagenome (meta-large) parameter *preset*, a *k-min* of 27 and with iterative increments of k (k-step) of 10 (exact parameters: –*presets meta-large –k-min 27 –k-step* 10 –*t* 20 –*m* 400e9). A total of 3,495,176,470 reads were assembled into 21,731,098 contigs (smallest contig size: 200 bases, largest: 489,715 bases, $N_{50}$: 681 bases, average contig size: 648 bases). Taxonomic annotation of the assembled contigs was performed by using *blastn* (November 19, 2020) matches against the NCBI "nt" database using an E value cutoff of 1e−4 (other parameters were at default values). Matches above 99% identity were assigned the reported taxonomic identity. Sequences with no

matches above the identity threshold were assigned taxonomy using a lowest common ancestor method generated on the top 50 *blastn* matches obtained. This method is closely inspired from the LCA algorithm implemented in MEGAN (82). Functional annotation of contigs was made using the following steps. First, the predicted proteins were determined from the contig nucleotide sequences using *ORFM* (default parameters: –*m* 96). A total of 233,744,501 proteins were predicted from open reading frames (ORFs). Then, the predicted protein sequences were annotated using Diamond's implementations for *blastp* (with the –*sensitive* parameter) against the SwissProt-UniProt database (November 24, 2020) in order to obtain gene ontology (GO) and KEGG ontology (KO) information. Finally, a database combining the taxonomic and functional information was made to be used as a reference for subsequent steps. The functional repertory (i.e., list of all KEGG pathways) in each sample was inferred from this database, at the most precise KEGG orthology level. Then, a table comprising the abundance of all pathways in each sample was produced and could be handled in the same way as standard ASV tables for downstream analyses. The functional reference database constructed is freely and publicly available on OSF (https://osf.io/dz6vf/). Details on the shotgun metagenomic database construction can be found in the flowchart of Fig. S8.

**Statistical analysis.** First, we aimed to understand to what extent environmental conditions drive the phylogenetic structure of global and transcriptionally active bacterial communities, and the inferred functional repertory (i.e., the list of all KEGG pathways) of these Amazonian bacterioplankton communities. We first computed the Shannon H diversity index using *plot_richness* from R *phyloseq* package (Fig. S3) and described the relative abundance of the different bacterial phyla in stacked bar plots from *ggplot2* (Fig. S1) in each water type. Then, we computed distance-based redundancy analyses (RDA) using *capscale* from *vegan* R package (84) to summarize the variation in the response variables (i.e., bacterioplankton communities) explained by environmental parameters (explanatory variables) (Fig. 2). Relative abundance (sum normalization) tables of ASVs were used as inputs for RDA analyses. We checked for multicollinearity by measuring variance inflation factors (VIF) using *vif.cca* from *vegan* (85) and then by computing stepwise variable selection using *ordistep* from *vegan* (86). The proportion of the variance explained by each variable was checked using *envfit*, also from *vegan* (86). Only environmental parameters with VIF < 10 and selected by *ordistep* were kept for RDAs (Fig. 2).

Then, we aimed to identify which taxa were associated with the variations between water types. We used a machine-learning approach to identify the taxa that best discriminated the different water types. To do so, we implemented Breiman's random forest algorithm for classification (87) using *randomForest* (88) with ntree = 500. This algorithm splits data in a training and a test set; the training set is used to construct consensus trees of classification via bootstrapping, and the test set (≈37% of the samples) is then used to estimate the node error rates in the trees of classification (i.e., the out-of-bag estimate of error). We isolated the 40 ASVs responsible for the most important mean decrease in GINI coefficient (measure of node purity) with significant *P* values following Bonferroni correction. These ASVs comprised the 40 taxa that best discriminated the different water types (with the lowest classification error rate) in the random forest tests. The relative abundance of these taxa in global and transcriptionally active bacterial communities was represented on heat maps (Fig. 3). The same approach was used to detect the inferred functions that best discriminated water types (Fig. S7).

We then characterized the FDOM (i.e., assessed the presence of its different humic-, fulvic-, and protein-like fractions) in the sampled environments using principal components analysis (PCA) (Fig. 4a), PARAFAC model components (Fig. 4b), and FEEM scans (Fig. S5). To test our hypothesis concerning the potential role of humic-like materials in structuring bacterioplankton communities, we assessed the correlation between the relative abundance of humic FDOM and the structures of global and transcriptionally active bacterioplankton communities. We first fitted the humic FDOM relative abundance on RDAs (Fig. 5a and b). Then, we assessed the Spearman correlations between the relative abundances of humic FDOM and bacterial ASVs (Fig. 5c and d). Correlations > 0.5 with *P* value < 0.05 (after Bonferroni correction) were plotted on Cytoscape v.3.7.1. Then, we used the same approach (i.e., Spearman correlations) at a higher taxonomic level, to identify bacterial genera of which the abundance correlated significantly with humic FDOM (Fig. 6). Based on these correlation results, we also investigated whether these genera possessed pathways involved in the degradation of humic DOM. To do so, we first identified in the literature (21, 58, 68) the potential enzymes currently known to be involved in humic FDOM degradation in bacteria (see list in Table S3). Then, we searched for the presence of these enzymes in the inferred functions of the genera correlated with the relative abundance of humic FDOM. We plotted these genera and their functional pathways using a heat map (Fig. 6).

**Data availability.** The data sets generated and analyzed during the current study can be found in the Sequence Read Archive (SRA) repository under BioProjectIDs PRJNA736442 and PRJNA736450. The accession numbers of the 90 metagenomes used to build the custom database are SRR1182511, SRR1182512, SRR1183643, SRR1183650, SRR1185413, SRR1185414, SRR1186214, SRR1199270, SRR1199271, SRR1199272, SRR1202081, SRR1202089, SRR1202090, SRR1202091, SRR1202095, SRR1204580, SRR1204581, SRR1205250, SRR1205251, SRR1205252, SRR1205253, SRR1209976, SRR1209977, SRR1209978, SRR1514963, SRR1515032, SRR1518285, SRR1522964, SRR1522971, SRR1522973, SRR1522974, SRR1786279, SRR1786281, SRR1786608, SRR1786616, SRR1787940, SRR1787943, SRR1788318, SRR1790487, SRR1790489, SRR1790644, SRR1790646, SRR1790647, SRR1790676, SRR1790678, SRR1790679, SRR1790680, SRR1792674, SRR1792852, SRR1793861, SRR1793862, SRR1796116, SRR1796118, SRR1796234, SRR1796236, SRR4831644, SRR4831645, SRR4831646, SRR4831647, SRR4831648, SRR4831649, SRR4831650, SRR4831651, SRR4831652, SRR4831653, SRR4831654, SRR4831655, SRR4831656, SRR4831657, SRR4831658, SRR4831660, SRR4831661, SRR4831662, SRR4831663, SRR4831664, SRR4831665, SRR4831666, SRR4831667, SRR4833053, SRR4833055, SRR4833056, SRR4833057, SRR4833059, SRR4833060, SRR4833062, SRR4833064, SRR4833067, SRR4833073, SRR4833077, SRR4833080, SRR4833081, SRR4833084, SRR4833086, SRR4833087, SRR4833089, SRR5123271, SRR5123272,

SRR5123273, SRR5123274, SRR5123275, SRR5123276, and SRR5123277. The scripts used for the 16S DNA/RNA sequence analysis, the input files including all metadata, the functional inference database, and the raw EEMS and absorbance scans data are all freely available on the Open Science Network platform (https://osf.io/dz6vf/). The main script for statistical analysis, an RData file containing the phyloseq objects, and the main KEGG table used were all uploaded as supplemental material.

## SUPPLEMENTAL MATERIAL

Supplemental material is available online only.

**SUPPLEMENTAL FILE 1**, PDF file, 2.6 MB.
**SUPPLEMENTAL FILE 2**, TXT file, 0.1 MB.
**SUPPLEMENTAL FILE 3**, TXT file, 2.9 MB.
**SUPPLEMENTAL FILE 4**, CSV file, 4.6 MB.
**SUPPLEMENTAL FILE 5**, CSV file, 3.6 MB.

## ACKNOWLEDGMENTS

This work was supported by the National Geographic Society, Natural Sciences and Engineering Research Council of Canada (NSERC), MITACS, and Ressources Aquatiques Québec through travel and field work grants (F.-É.S.). This study was also supported in part by a NSERC Discovery grant (N.D.), the Instituto Nacional de Ciência e Tecnologia de Adaptações da Biota Aquática da Amazônia (INCT ADAPTA) project (A.L.V.), a Canada-Brazil Awards Joint Research Project (N.D. and A.L.V.), and by funds from Conselho Nacional de Desenvolvimento Cientifico e Technologico (CNPq), Fundação de Amparo à Pesquisa do Estado do Amazonas (FAPEAM), and Coordenação de Aperfeiçoamento de Pessoal de Nível Superior (CAPES).

We thank Thiago Nascimento, Reginaldo Oliveira, and Nazaré Paula for technical support with field work logistics. We thank Roxanne Dhommée for support in the molecular biology laboratory work.

We declare no conflict of interest.

F.-É.S., A.L.V., and N.D. designed the study; F.-É.S., N.L., A.H., and N.D. performed field sampling; F.-É.S., N.L., and P.-L.M. conducted RNA extractions and prepared 16S rRNA libraries; F.-É.S., N.L., and S.B. conducted the bioinformatical and statistical analyses; F.-É.S. wrote the manuscript; all authors revised the manuscript.

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
