## [Reviewer comments · Microbiology Spectrum]

Microbiology Spectrum

Bacterioplankton communities in dissolved organic carbon-rich Amazonian black water.

Francois-Etienne Sylvain, Sidki Bouslama, Aleicia Holland, Nicolas Leroux, Pierre-Luc Mercier, Adalberto Luis Val, and Nicolas Derome

Corresponding Author(s): Francois-Etienne Sylvain, Universite Laval

Review Timeline:

Submission Date:	November 22, 2022
Editorial Decision:	December 8, 2022
Revision Received:	February 20, 2023
Editorial Decision:	May 3, 2023
Revision Received:	May 3, 2023
Accepted:	May 4, 2023

Editor: Eva Sonnenschein

Reviewer(s): Disclosure of reviewer identity is with reference to reviewer comments included in decision letter(s). The following individuals involved in review of your submission have agreed to reveal their identity: Elias Broman (Reviewer #1); Pengfa Li (Reviewer #2)

Transaction Report:

DOI: <https://doi.org/10.1128/spectrum.04793-22>

December 8, 2022

Mx. Francois-Etienne Sylvain
Universite Laval
Biology
1030 avenue de la Médecine
Quebec, QC
Canada

Re: Spectrum04793-22 (Bacterioplankton communities in dissolved organic carbon-rich Amazonian black water.)

Dear Mx. Francois-Etienne Sylvain:

Link Not Available

Sincerely,

Eva Sonnenschein

Journals Department
Reviewer comments:

Reviewer #1 (Comments for the Author):

In the manuscript "Bacterioplankton communities in dissolved organic carbon rich Amazonian black water" by Dr. Sylvain investigates prokaryotic communities in three different water types of the Amazon River. In addition, they measured several environmental variables and characterized the organic carbon using absorbance and fluorescence-based methods. The manuscript has previously been submitted to the journal Limnology and Oceanography and this is the revised version.

The manuscript contains a large dataset of environmental variables and 16S rRNA gene and 16S rRNA amplicon data and merits publication. After the revision of the manuscript it has been restructured and the text updated. However, I still found that

the structure can be improved (I would suggest to first present the sites with the map, then environmental conditions, and then the microbial data, and finally the relationship between the two datasets). I also think that some of the previous reviewer comments were not sufficiently addressed. The authors decided to not fully follow the suggested steps in the DADA2 pipeline and instead do a blast search with a criterion of 99 % identity of their ASVs and implement an LCA algorithm. This seems to have classified many of their ASVs on a high taxonomic level such as unclassified bacteria or phyla. Considering it will probably only take 1 or 2 days to reclassify the ASVs against a curated 16S rRNA gene database such as SILVA, following the DADA2 pipeline, I do not see a reason why the authors cannot do this. This procedure will likely classify most of the ASVs down to genus level and can be used as a comparison to their current dataset. If the datasets are similar this will further strengthen their current classified data now analyzed in the manuscript.

The authors focus a lot of the results and discussion on Polynucleobacter which was found to correlate with humic FDOM. However, how significant is Polynucleobacter in the microbial communities? I could not find information of how much % they represent of the whole prokaryotic community? Or their difference in % between sites. A large part of the manuscript might be focused on a very small portion of the microbial community. This is not necessarily wrong, and if this is the case I suggest to include Polynucleobacter in the aim or title. After SILVA classification it is likely that Polynucleobacter will still be associated with FDOM, however the SILVA classifications might help to label the genus of some of the ASVs now classified as "Bacteria" which correlated with FDOM.

It is also not clear how exactly the metagenomic data was analysed. The authors write that they did a "16S approach" but what does this mean? In the dataset the authors already had 16S rRNA gene and 16S rRNA amplicons so why focus the metagenome analyses on 16S rRNA genes? It also seems that after assembly of contigs the quality trimmed reads were not mapped on the assembly so I am not sure how the authors were able to get information on counts per functional gene? See more detailed comments on this below.

There is no reason that the bioinformatic methods are in the supplementary methods. This information needs to be moved to the methods in the manuscript.

Additional comments:

I think the introduction is missing a small section on previous microbial work done in the study area (or mentioning the lack of).

Line 114-117: Here the authors extracted RNA, synthesized cDNA, and transcribed a partial fragment of ribosomal molecules (rRNA). This is not the same as RNA transcripts which is mRNA transcribed from genes. The phrase "transcriptional activity" is therefore not directly what the authors analyzed. I suggest to remove this wording here and elsewhere in the manuscript. I suggest the authors to instead call these analyzes "16S rRNA gene" and "16S rRNA" in the text and figures.

I am not sure why the headers in the results state the objectives. The objectives should be stated in the aim. I also think the text can be improved to answer the objectives more clearly. For example, there is no text mentioning which environmental variables shaped the composition of the rRNA gene or rRNA communities, or the functional gene composition. Also, considering the authors found Polynucleobacter to be one of the important taxa, the text or figures do not show how much % these bacteria represented of the community, or what the difference in % was between water types.

124-126: This is discussion, and the findings in reference 32 might be worthwhile to also mention in the introduction if it's a previous study conducted in the same study area.

Lines 126-127: What do the authors mean with "seemed largely unaffected by", was it statistically significant?

Lines 128-129: "NDMS analyses based on the taxonomic structure". What kind of beta diversity index was used, was it e.g. Bray-Curtis?

Lines 135-136: This is discussion.

Lines 140: This can be more informative. What were these FDOM components that were associated with the taxonomic communities?

Line 142: How many permutations were done for the PERMANOVA analysis?

Line 147: change "have shown" to "showed". Also "Betadisper permutests" are referred to as "they have shown" in the text which should be changed to "The tests showed"

Lines 161-178: Here I recommend the authors to follow the DADA2 pipeline and classify the ASVs against the SILVA SSU database and compare the results to their NCBI classifications. A good example would be to compare supp. Figure 1 with the SILVA taxonomy. And a heatmap of the top most abundant genera (e.g. with a >0.5% threshold). Considering the primers used the classifications against SILVA are likely to be done on the genus level. The text is also missing numbers, such as relative

abundance (%) to be able to compare between the water types when reading the text.

Line 208: This needs to be better explained. What does the metagenome database consist of? Annotated functional genes and their taxonomy or only functional gene annotations? What classification was done for the taxonomy? How did you get counts per gene if you did not map the reads on the assembled contigs (as this was not mentioned in the methods)?

Lines 218-219: Here and at most other locations in the manuscript, statements are made with no numbers or statistics to back it up. What was the correlation and was it significant?

Lines 230-236: I do not understand this. What is "metagenomics for a 16S-based approach"?

Line 249: Sequencing rRNA is not the same as measuring transcriptional activity. here the authors would need to do RNA seq and investigate mRNA transcripts. Here the authors can rephrase the sentence to say they investigated microbial communities by targeting both the 16S rRNA gene and 16S rRNA.

Line 260-262: Which environmental parameters were most associated with the communities? This needs to be clarified here.

Lines 264-266: the RDA seems to show that more % on axis 1 and 2 are explained in the 16S rRNA gene than 16S rRNA data (Fig. 2). This is interesting and not mentioned in the results.

Line 274-284: I am not sure why this is being discussed here, as there is no mentioning in the discussion of which relevant functional genes were present in the water bodies or associated with the environmental conditions.

Line 278: I do not follow what is meant here, didn't the authors classify functional genes from shotgun sequencing?

Lines 301-306: Is it possible that some bacteria associated with FDOM are derived from terrestrial organic matter leaching into the riverine water?

Lines 373-379: I don't think this is really needed in the conclusions, as this mostly repeats the methods.

Lines 383-384: Remove repetition of methods

Line 404: What is this Nucleic Acid Preservation buffer? Write the contents in the text. Will it keep RNA stable?

Line 449: Shouldn't there be a reference for EPA Method 160.2?

Line 489-496: Most journals accept supplementary data files and I think the authors should utilize this and not solely rely on an external service (which can be both edited and removed after publication). ASV tables, KEGG tables, scripts etc. can all be uploaded as supplementary files accompanying the manuscript.

Lines 500-501: Did the RNA extraction include a DNase treatment to remove DNA contamination?

Line 513: Details on the methods for multiplexing with indexes are missing. This information is needed to be able to reproduce the study.

Lines 517-18: There is no reason to keep the 16S bioinformatics in the supplementary methods. Move to the main text.

Line 521: phyloseq package version?

Line 523-530: The metagenome bioinformatics supplementary methods should be moved to the main text. Some brief details on the additional metagenome samples would also be useful, such as what type of waters were sampled (black, white, clear)? Was it surface or bottom water?

Lines 532-547: This section describes what statistical tools were used, but not how it was done. What R packages were used? Or other software? Was the data normalized (e.g. rarified) before alpha diversity analysis?

Line 536: Is it the Shannon's H diversity index?

Line 538-545: Was the RDA and NMDS based on ASV level or e.g. genus level? Was the data normalized before these analyses (e.g. relative abundances)?

Line 546: How come only unweighted unifracs distances were used? This will only consider presence and absence information, and not the relative abundance of the ASVs.

Lines 572: Should it be > 0.5 and < -0.5 ?

Lines 572-576: Was any add-on used with Cytoscape? E.g. CoNet?

Lines 585: I think "We thank two anonymous reviewers for their essential input on the original version of this manuscript." is no longer relevant in the acknowledgements.

Table 2: Here it would be useful to also include a column of "Water color" as in Table 1.

Figure 1: Shouldn't this caption have at least one reference where the information was retrieved from?

Figure 1: Oxydation is a French word. Should it be Oxidation?

Figure 2: "inferred functional repertory". Is this based on the functional annotation of the metagenomes or the taxonomy of the amplicon/metagenome data?

Figure 3: The relative abundance scale does not have any numbers. How much does the scale represent, is it 0-100%?

Figure 5: I think that the correlation network is not very informative. Typically each line will also show if it's a positive or negative correlation, here instead all lines are grey or colored based on taxonomy. Some lines are also black and I am not sure what this color denotes. The taxa is also only shown on a high taxonomic level. It would be good to double-check if SILVA annotations will give information on these unclassified taxa.

Figure 6: The color scale has no numbers and is only indicated by minus and plus symbols. What is yellow and red? Is it -1 to 1 Spearman's rho?

Figure 7: This should be figure 1

Comments on the supplementary methods-----

What settings were used with the different DADA2 steps, such as mergePairs?. For example, were default settings used? This information will make it possible to reproduce the study.

What settings were used with blastn?

What was the date of the ncbi database?

What is the 16S Microbial database exactly?

The authors write: "This method is closely inspired from the LCA algorithm implemented in MEGAN (Huson et al. 2016)." But they never explain what was done. It seems this method had a 99% identity threshold and might have caused many classifications to be ranked on a higher taxonomic level, with many ASVs also being classified as "Bacteria". I suggest to also classify the ASVs sequences against SILVA and use this as a comparison.

What trimmomatic data was used? Was it the filtered paired, filtered unpaired, or both? This information is required to be able to reproduce the study.

Was each sample assembled using megahit individually or was a co-assembly conducted? This needs to be clarified.

The authors mention that a table comprising the abundance of pathways in each sample was produced. However, the authors mention no methods on mapping reads onto the assembly so how did the authors get information on gene abundances?

Reviewer #2 (Comments for the Author):

Sylvain et al. investigated the bacterioplankton communities in Amazonian black water. Given the importance of Amazonian River in global ecosystem services, I believe the data presented in the current study are valuable. Since this is an easy study, I do not have many concerns. All displays are eye-catching. However, I think the paper is not well organized and written. I would suggest the readers to read more related papers and re-organize this paper. Below I listed some examples.

Abstract

In the beginning of the Abstract, I see no knowledge gap. Why this study should be conducted? Just because some phenomenon remain unsolved? Billions of things remain unsolved, but we needn't, and it's impossible, to investigate every

UNSOLVED thing.

Line 39-41: This statement is too absolute. First, the MAGs were assembled from the literature, not your REAL data. Second, they may have these genes, but having genes does not necessarily function.

Introduction:

The authors gave two hypotheses in Introduction. Hypothesis 1 looks fine, but I would suggest changing it like 'Given that bacterioplankton communities have been demonstrated to involve in the transformation of DOM, we hypothesize that...'. If you propose this hypothesis just because 'as shown in other ecosystems', I may disagree because your ecosystem looks highly different from others.

Line 81: A little confusing to me. Now that Rio Negro contains high DOM, why is it oligotrophic?

Results

I have to say that I feel very tired to read the Results part, although, as I stated above, this is a very easy and small study. Let's look at the first paragraph of Results. The first sentence, it's not necessary to say 'The Amazonian bacterioplankton was similar in composition to what has been reported in (32)'. You just show your results, and discuss and compare your results with others in Discussion. Why start discussing in the first sentence of Results! In Line 126, the authors stated 'While Shannon alpha diversity seemed largely unaffected by water colors'. Is this really a valid result? What valid information can we get from this? Why not show your data? In Objective #1, the authors aimed to identify the environmental variables shaping the composition, transcriptional activity and inferred functions of Amazonian bacterioplankton. But in the main text, I don't see any specific variables. So, the authors, actually, failed to tell us anything in this long paragraph. WHY NOT SHOW YOUR DATA IN RESULTS? This is an important problem across the whole paper.

Line 195: I've never heard about 'Co-abundance network'. Is this a newly invented word?

Discussion:

Please try to shorten it. It's too long and boring.

Others:

Line 278 : PICRUST should be PICRUST

Line 552: ntree = 100 is not a commonly used parameter. We commonly use 500. Using 100 can save time but increase error. Your data are not very large, why use 100?

Staff Comments:

Preparing Revision Guidelines

Please return the manuscript within 60 days; if you cannot complete the modification within this time period, please contact me. If you do not wish to modify the manuscript and prefer to submit it to another journal, please notify me of your decision immediately so that the manuscript may be formally withdrawn from consideration by Microbiology Spectrum.

Corresponding authors may join or renew ASM membership to obtain discounts on publication fees. Need to upgrade your

membership level? Please contact Customer Service at Service@asmusa.org.

Response to reviewers – Manuscript #: Spectrum04793-22R1

Reviewer #1

Comment: In the manuscript "Bacterioplankton communities in dissolved organic carbon rich Amazonian black water" by Dr. Sylvain investigates prokaryotic communities in three different water types of the Amazon River. In addition, they measured several environmental variables and characterized the organic carbon using absorbance and fluorescence-based methods. The manuscript has previously been submitted to the journal *Limnology and Oceanography* and this is the revised version.

The manuscript contains a large dataset of environmental variables and 16S rRNA gene and 16S rRNA amplicon data and merits publication.

Response: Thank you for this comment!

After the revision of the manuscript it has been restructured and the text updated. However, I still found that the structure can be improved (I would suggest to first present the sites with the map, then environmental conditions, and then the microbial data, and finally the relationship between the two datasets). I also think that some of the previous reviewer comments were not sufficiently addressed. The authors decided to not fully follow the suggested steps in the DADA2 pipeline and instead do a blast search with a criterion of 99 % identity of their ASVs and implement an LCA algorithm. This seems to have classified many of their ASVs on a high taxonomic level such as unclassified bacteria or phyla. Considering it will probably only take 1 or 2 days to reclassify the ASVs against a curated 16S rRNA gene database such as SILVA, following the DADA2 pipeline, I do not see a reason why the authors cannot do this. This procedure will likely classify most of the ASVs down to genus level and can be used as a comparison to their current dataset. If the datasets are similar this will further strengthen their current classified data now analyzed in the manuscript.

Response: In our study, the taxonomic annotation of ASVs consisted of two steps using NCBI databases. Firstly, *blastn* queries were conducted against NCBI "16S Refseq Nucleotide database". This database is the most exhaustive manually curated 16S rRNA gene database currently available, and uses over 150 sources, such as the *Catalog of Life*, the *Encyclopedia of Life*, *NameBank* and *WikiSpecies*. When a match above 99% identity was obtained after the *blastn* query, ASVs were assigned their reported taxonomic identity. Secondly, the sequences with no matches above the identity threshold were assigned taxonomy using a lowest common ancestor (LCA) method generated on the top 50 *blastn* matches obtained on that same NCBI database.

The use of the NCBI database for this project, rather than other databases, aimed to achieve three goals: **(1)** Prevent ambiguous annotations; **(2)** Use the most up-to-date data available; and **(3)** Optimize the match between 16S and shotgun sequencing datasets used in this study.

(1) Preventing ambiguous annotations: Our divergence from the annotation method suggested in the DADA2 documentation was intentional and based on published findings by Edgar (2018) and our own past experiences with using Naive Bayes classifiers on SILVA and RDP databases. By default, the behaviour of the "addSpecies" method in DADA2 will output one match/taxonomic identity per query if unambiguous. This can be remedied by using the "allowMultiple=True" option which will output multiple species level matches separated by a "/" character. Studies involving subjects that are less studied and less represented in databases

such as SILVA and analyzed using the suggested DADA2 methodology will often report results that showcase a far better species/genus level discrimination than what short read 16S amplicon methods can actually offer, since there are less chances for ambiguous multiple matches per query. Furthermore, the paper from Edgar (2018) shows a high level of conflicting annotations reported on SILVA and RDP databases, which we have found to be even more apparent on datasets of environmental and less studied origins.

(2) Using the most up-to-date data available: NCBI taxonomic classification files are updated on a daily basis (Balvočiūtė and Huson 2017), while other databases such as SILVA, RDP or Greengenes are updated much less frequently. With a more up to date database such as NCBI, multiple high identity and coverage hits on phylogenetically distant strains can result in annotations at the Kingdom/Phylum levels. In databases with less frequent updates, a lack of diversity can produce artificially phylogenetically close matches to query sequences which result in a seemingly more resolved taxonomic classification, but which is less representative of the “real” taxonomic complexity.

(3) Optimizing the match between 16S rRNA and shotgun sequencing datasets: Furthermore, our approach maximized the match between 16S rRNA sequences and the shotgun sequencing datasets that were used to infer the functional repertory of bacterial communities. Indeed, an article from Balvočiūtė and Huson (2017), focused on the comparison of the resolution of the four main taxonomic databases (SILVA, RDP, Greengenes and NCBI), concludes on the following:

“Therefore, we recommend using the NCBI taxonomy as a common framework when comparing analyses performed on different taxonomic classifications. While the SILVA taxonomy is widely used for 16S studies, one should consider using the NCBI taxonomy in studies that use both targeted 16S sequencing and shotgun sequencing.”

Overall, we have weighed the advantages and disadvantages of various annotation methodologies available to us and have decided that the lower annotation error rates, the frequent updates, and the exhaustive nature of the NCBI 16S Refseq database made this database the best taxonomic tool for this study.

References

Balvočiūtė, M., Huson, D.H. SILVA, RDP, Greengenes, NCBI and OTT — how do these taxonomies compare?. 2017. BMC Genomics 18 (114). doi: 10.1186/s12864-017-3501-4

Edgar, R. 2018. Taxonomy annotation and guide tree errors in 16S rRNA databases. PeerJ 6:e5030. doi: 10.7717/peerj.5030

Comment: The authors focuses a lot of the results and discussion on Polynucleobacter which was found to correlate with humic FDOM. However, how significant is Polynucleobacter in the microbial communities? I could not find information of how much % they represent of the whole prokaryotic community? Or their difference in % between sites. A large part the manuscript might be focused on a very small portion of the microbial community. This is not necessary wrong, and if this is the case I suggest to include Polynucleobacter in the aim or title. After SILVA classification it is likely that Polynucleobacter will still be associated with FDOM, however the SILVA classifications might help to label the genus of some of the ASVs now classified as "Bacteria" which correlated with FDOM.

Response: Indeed, *Polynucleobacter* was one of the key taxa in this study, as we found that among all taxa, *Polynucleobacter* was the one showing the strongest correlation to humic DOM, and its inferred functional repertory suggested it possessed several genes involved in humic DOM degradation. However, this taxon was relatively not abundant (mean abundance around 0.05 to 2.25% depending on the water type). This information has been added to the Results of the revised manuscript (lines 249-251):

“Four main results suggest that among all taxa, the genus Polynucleobacter, which had a relatively low abundance (mean of 0.05-2.25 %) in global and transcriptionally-active bacterioplankton (Suppl. Fig. 6), showed the strongest association to humic FDOM.”

We also added a figure in Supplementary material (Suppl. Fig. 6) showing these relative abundance results. The following information was also added to the Abstract of the revised manuscript (lines 38-41):

“The strongest correlations were found in Polynucleobacter, Methylobacterium and Acinetobacter genera, three low abundant but omnipresent taxa which possessed several genes involved in the main steps of the β -aryl ether enzymatic degradation pathway of diaryl humic DOM residues.”

The importance of *Polynucleobacter* is stated in the Abstract, Results, Discussion and Conclusion, however, we prefer not to include it in the title, as we studied the bacterioplankton community as a whole without targeting a specific taxon from the beginning of the project.

Comment: There is no reason that the bioinformatic methods are in the supplementary methods. This information needs to be moved to the methods in the manuscript.

Response: The bioinformatic methods have been moved to the Methods of the manuscript. See sections **16S rRNA sequence analysis** (lines 513-579) and **Shotgun metagenome functional database** (lines 581-618).

Additional comments:

Comment: I think the introduction is missing a small section on previous microbial work done in the study area (or mentioning the lack of).

Response: The following paragraph has been added to the Introduction of the revised manuscript to clarify the gaps in the current Amazonian bacterioplankton literature that have not been addressed by similar studies in the area (lines 100-111):

“ Despite its relevance for global-scale elemental cycling and primary production processes, there is a limited understanding of the taxonomical and functional structure of the Amazon River bacterioplankton. A few studies have focused on these bacterial communities, however most of them did not sample in different water types (21-29). Several of the aforementioned studies also included a limited number of habitats sampled (e.g. only one site in (23)), or were focused on the dynamics of bacterioplankton in the plume downstream of the Amazonian River (25-27) rather than the communities in upstream black water systems per se. A previous study (21) found that the genera Ramlibacter, Planktophilia, Methylopumilus, Limnohabitans and Polynucleobacter were enriched in DOM degradation pathways along the Amazon River, however the abundance of these genera in accordance to humic DOM and whether they are transcriptionally-active or not remains unknown.”

Comment: Line 114-117: Here the authors extracted RNA, synthesized cDNA, and transcribed a partial fragment of ribosomal molecules (rRNA). This is not the same as RNA transcripts which is mRNA transcribed from genes. The phrase "transcriptional activity" is therefore not directly what the authors analyzed. I suggest to remove this wording here and elsewhere in the manuscript. I suggest the authors to instead call these analyzes "16S rRNA gene" and "16S rRNA" in the text and figures.

Response: We agree with the reviewer that it is important to make clear that our study did not analyze mRNA, but rRNA from RNA extracts. This approach enabled us to study the taxonomic structure of the transcriptionally-active bacterioplankton community. In the revised manuscript, we changed all instances of the wording "the transcriptional activity" to "the taxonomic structure of the **transcriptionally-active** bacterioplankton community" (e.g. see at lines 129, 149 and 153). In contrast, when referring to community characterized from rRNA gene analyzes (from DNA extracts) we used the wording "the **global** bacterioplankton community".

The Figures 2, 3 and 5 were also modified accordingly.

Comment: I am not sure why the headers in the results state the objectives. The objectives should be stated in the aim.

Response: In the revised manuscript, we removed the headers of the results which stated the objectives, as the objectives are already defined in the last paragraph of the Introduction (lines 113-124):

" In this study, we aimed to identify the most important environmental variables shaping the Amazonian bacterioplankton community structure and inferred functional profile. Given that bacterioplankton communities have been shown to be involved in the transformation of DOM (30-32), we hypothesized that [...]. Secondly, we aimed to better understand potential interactions between humic FDOM and the Amazonian bacterioplankton. We hypothesized that [...]"

Comment: I also think the text can be improved to answers the objectives more clearly. For example, there is no text mentioning which environmental variables shaped the composition of the rRNA gene or rRNA communities, or the functional gene composition.

The following text was added to the Results section of the revised manuscript to detail which environmental variables shaped the taxonomic composition and functional profiles of the bacterioplankton communities (lines 152-168):

"The environmental parameters that significantly influenced the taxonomic structure of global bacterioplankton, of transcriptionally-active bacterioplankton, and the inferred functional repertory of bacterioplankton were not identical. The taxonomic structure of global bacterioplankton from black water was mostly driven by the concentration of Cd^{2+} , Co^{2+} , humic DOC, fluvic DOC, total DOC and levels of SAC340, but in white and clear waters it was associated to the concentration of Mg^{2+} , Na^+ , and the levels of pH, conductivity and Abs254/365. The taxonomic structure of transcriptionally-active bacterioplankton in black water was driven by the concentration of Co^{2+} , Pb^{2+} , and Cd^{2+} , but in white and clear waters the main drivers were the concentrations of Ca^{2+} , Mg^{2+} , K^+ , protein-like DOC, fluvic DOC and the level of Abs254/365. The inferred functional repertory from blackwater was associated by the concentrations of Fe^{3+} , Mn^{2+} and Cd^{2+} , while in white and clear waters it was affected by the

concentration of Ca^{2+} , silicate and the levels of pH and Abs254/365. Overall, the taxonomic structures and inferred functional repertory were driven by several parameters associated with the relative abundance of the different FDOM components (i.e. the relative abundance of humic, flavic and protein-like DOC, in addition to the SAC340 and Abs254/365 ratios), which appear in red in RDAs of Fig. 2.”

Comment: Also, considering the authors found *Polynucleobacter* to be one of the important taxa, the text or figures do not show how much % these bacteria represented of the community, or what the difference in % was between water types.

Response: Indeed the genus *Polynucleobacter* is an important taxa in this study, as it is the genus which relative abundance correlated the most with the abundance of humic FDOM. Furthermore, the inferred functional repertory of this taxa included genes involved in all main steps of humic compounds degradation. However, the ASVs from this genus represented a relatively low proportion of the bacterioplankton communities investigated (average abundances of 0.05-2.25%). In the revised manuscript, the relative abundance of *Polynucleobacter* ASVs is displayed in a new Supplementary figure (Suppl. Fig. 6). Furthermore, the following sentence has been modified (see bold text) in the Results section (lines 249-251):

“Four main results suggest that among all taxa, the genus *Polynucleobacter*, **which had a relatively low abundance (mean of 0.05-2.25 %) in global and transcriptionally-active bacterioplankton (Suppl. Fig. 6)**, showed the strongest association to humic FDOM.”

See Suppl. Fig. 6 below:

Suppl. Figure 8: Average relative abundance of *Polynucleobacter* ASVs in each water type for global bacterioplankton (a) and for transcriptionally-active bacterioplankton (b).

Comment: 124-126: This is discussion, and the findings in reference 32 might be worthwhile to also mention in the introduction if it's a previous study conducted in the same study area.

Response: We removed this sentence from the Results section and included the information (in bold) in the Discussion of the revised manuscript (lines 272-276):

“Our results showed **that the Amazonian bacterioplankton was similar in composition to what has been reported in (22)**, with an important influence of water type on the taxonomic

structure of global and transcriptionally-active bacterioplankton communities, and on their functional profiles (Fig. 2)."

Comment: Lines 126-127: What do the authors mean with "seemed largely unaffected by", was it statistically significant?

Response: Shannon alpha diversity was not significantly affected by water types. This information was added to the revised manuscript (lines 137-139):

"While Shannon alpha diversity did not significantly differ between water types ($p > 0.05$, mean Shannon diversity between 6-7 for all water types, see Suppl. Fig. 2, 3, 4), beta diversity analyses showed that [...]"

Comment: Lines 128-129: "NDMS analyses based on the taxonomic structure". What kind of beta diversity index was used, was it e.g. Bray-Curtis?

Response: These NMDS analyses were removed from the revised manuscript as they were redundant with the RDA ordination analysis found in the main text.

Comment: Lines 135-136: This is discussion.

Response: This sentence was removed from the Results section of the revised manuscript.

Comment: Lines 140: This can be more informative. What were these FDOM components that were associated with the taxonomic communities?

Response: The details (**in bold**) have been added to this sentence in the revised manuscript (see lines 164-168):

*"Overall, the taxonomic structures and inferred functional repertory were driven by several parameters associated with the relative abundance of the different FDOM components (**i.e. the relative abundance of humic, fluvic and protein-like DOC, in addition to the SAC340 and Abs254/365 ratios**), which appear in red in RDAs of Fig. 2."*

Comment: Line 142: How many permutations were done for the PERMANOVA analysis?

Response: The following information (**in bold**) has been added to this sentence in the revised manuscript (lines 170-171):

*"PERMANOVA analyzes (**999 permutations**) have shown that water type was significantly associated with the taxonomic structure [...]"*

Comment: Line 147: change "have shown" to "showed". Also "Betadisper permutests" are referred to as "they have shown" in the text which should be changed to "The tests showed"

Response: These sentences have been modified (see bold text) as suggested (lines 170-177):

*"PERMANOVA analyzes (999 permutations) **have shown** that water type was significantly associated with the taxonomic structure of global bacterioplankton [...]. Betadisper permutests (1000 permutations) **showed** homogenous variance between groups for the inferred functions dataset ($F = 0.73$, $df.res = 167$, $p = 0.45$)."*

Comment: Lines 161-178: Here I recommend the authors to follow the DADA2 pipeline and classify the ASVs against the SILVA SSU database and compare the results to their NCBI classifications. A good example would be to compare supp. Figure 1 with the SILVA taxonomy. And a heatmap of the top most abundant genera (e.g. with a >0.5% threshold). Considering the primers used the classifications against SILVA are likely to be done on the genus level. The text is also missing numbers, such as relative abundance (%) to be able to compare between the water types when reading the text.

Response: Please refer to prior justification for not using DADA2 methodology.

This comment also refers to the text associated to the results of Figure 3. In this figure, the lightness/darkness of each ASV x sample combination is scaled according to the abundance of the same ASV in all other samples. Thus, since the relative abundance is relative to what is found in other samples, and does not depend on the abundance of different ASVs from the same sample, it is impossible to establish a common scale with abundance values for all ASVs (all ASVs would need their own scale). This explains why (1) there is no numbers associated to the relative abundance scale/legend on Figure 3; and (2) the relative abundance values are not specified in the text.

Comment: Line 208: This needs to be better explained. What does the metagenome database consist of? Annotated functional genes and their taxonomy or only functional gene annotations? What classification was done for the taxonomy? How did you get counts per gene if you did not map the reads on the assembled contigs (as this was not mentioned in the methods)?

Response: The metagenome database is a repertoire of functions derived from predicted proteins translated from predicted ORFs mapped to the assembled metagenome. Genomic information from large assembled contigs were used to inform the taxonomic annotation fields of the database and the associated predicted functions obtained from protein matches (and homology) were used to inform the functional annotation fields. The goal was to obtain a repertoire of functions to be expected per-taxon observed. 16S rRNA amplicon counts were used as a signal for the relative abundance of the different taxa. Changes in the relative abundance of each ASV were then used to infer the impact on the global predicted functional repertoire.

Comment: Lines 218-219: Here and at most other locations in the manuscript, statements are made with no numbers or statistics to back it up. What was the correlation and was it significant?

Response: The sentence referred to in this comment was restructured to the following (lines 249-251):

“Four main results suggest that among all taxa, the genus Polynucleobacter, which had a relatively low abundance (mean of 0.05-2.25 %) in global and transcriptionally-active bacterioplankton (Suppl. Fig. 6), showed the strongest association to humic FDOM.”

We also carefully reviewed the Results section of the revised manuscript, to make sure that every time a statement is made on the general results, the statistical values that lead to this statement are specified. For instance see the first paragraph of the revised Results here below (the statistics added to the revised manuscript are in **bold**) (lines 136-150):

“ The Amazonian bacterioplankton showed a rich abundance of Proteobacteria, Actinobacteria, and Cyanobacteria (Suppl. Fig. 1). While Shannon alpha diversity did not significantly differ between water types ($p > 0.05$, mean Shannon diversity between 6-7 for all water types, see Suppl. Fig. 2, 3, 4), bêta diversity analyses showed that bacterioplankton communities significantly clustered according to water type in RDA analyses based on the taxonomic structure and inferred functions of these communities (RDA p-values < 0.001 , $F = 2.95-20.4$, see Fig. 2). These RDAs (Fig. 2) suggest that white and clear water communities were similar, and differed from black water communities. This result was confirmed by the error rates from the confusion matrix of the Random Forest classification: The rates of misclassification of clear water samples (0.08-0.58) were always higher between clear and white, than between clear and black water samples (0-0.31) (Suppl. Table 1). The RDAs on Fig. 2a,b show that a higher proportion of the total variance is explained by axes 1 and 2 for the global bacterioplankton (27.31%) than for the transcriptionally-active bacterioplankton (13.01%).”

We also added an additional table in Supplementary material (Suppl. Table 1) to detail the results of the Random-Forest classification tests, which were not included in the previous version of the manuscript.

Comment: Lines 230-236: I do not understand this. What is "metagenomics for a 16S-based approach"?

Response: The paragraph referred to in this comment has been removed from the revised manuscript.

Comment: Line 249: Sequencing rRNA is not the same as measuring transcriptional activity. here the authors would need to do RNA seq and investigate mRNA transcripts. Here the authors can rephrase the sentence to say they investigated microbial communities by targeting both the 16S rRNA gene and 16S rRNA.

Response: As mentioned in a previous response, we agree with the reviewer that it is important to make clear that our study did not analyze mRNA, but only rRNA from RNA extracts. This approach enabled us to study the taxonomic structure of the transcriptionally-active bacterioplankton community. In the revised manuscript, we changed all instances of the wording “the transcriptional activity” to “the taxonomic structure of the **transcriptionally-active** bacterioplankton community”. In contrast, when referring to community characterized from rRNA gene analyzes (from DNA extracts) we used the wording “the **global** bacterioplankton community”.

Comment: Line 260-262: Which environmental parameters were most associated with the communities? This needs to be clarified here.

Response: This sentence has been modified to the following (lines 276-280):

“At the taxonomic and functional levels, we showed that among the environmental parameters that were the most associated with community clustering (DOC quantity and type, pH, conductivity and concentrations of Cd^{2+} , Co^{2+} , Mg^{2+} , Na^+ , Ca^{2+} , K^+ , Pb^{2+} , Fe^{3+} , Mn^{2+} and silicates), several are known to be the main parameters driving differences between water types.”

The parameters referred to in this sentence are also detailed in Figure 2 and in the corresponding revised Results section (lines 152-168):

“ The environmental parameters that significantly influenced the taxonomic structure of global bacterioplankton, of transcriptionally-active bacterioplankton, and the inferred functional repertory of bacterioplankton were not identical. The taxonomic structure of global bacterioplankton from black water was mostly driven by the concentration of Cd^{2+} , Co^{2+} , humic DOC, fluvic DOC, total DOC and levels of SAC340, but in white and clear waters it was associated to the concentration of Mg^{2+} , Na^+ , and the levels of pH, conductivity and Abs254/365. The taxonomic structure of transcriptionally-active bacterioplankton in black water was driven by the concentration of Co^{2+} , Pb^{2+} , and Cd^{2+} , but in white and clear waters the main drivers were the concentrations of Ca^{2+} , Mg^{2+} , K^+ , protein-like DOC, fluvic DOC and the level of Abs254/365. The inferred functional repertory from blackwater was associated by the concentrations of Fe^{3+} , Mn^{2+} and Cd^{2+} , while in white and clear waters it was affected by the concentration of Ca^{2+} , silicate and the levels of pH and Abs254/365. Overall, the taxonomic structures and inferred functional repertory were driven by several parameters associated with the relative abundance of the different FDOM components (i.e. the relative abundance of humic, fluvic and protein-like DOC, in addition to the SAC340 and Abs254/365 ratios), which appear in red in RDAs of Fig. 2.”

Comment: Lines 264-266: the RDA seems to show that more % on axis 1 and 2 are explained in the 16S rRNA gene than 16S rRNA data (Fig. 2). This is interesting and not mentioned in the results.

Response: We agree with the reviewer that this is an interesting result! We added the following sentence to the Results section of the revised manuscript (lines 147-150):

“The RDAs on Fig. 2a,b show that a higher proportion of the total variance is explained by axes 1 and 2 for the global bacterioplankton (27.31%) than for the transcriptionally-active bacterioplankton (13.01%).”

Comment: Line 274-284: I am not sure why this is being discussed here, as there is no mentioning in the discussion of which relevant functional genes were present in the water bodies or associated with the environmental conditions.

Response: The reviewer is right: The Discussion does not focus on specific functional genes associated to different environmental conditions. The current manuscript only presents a general overview of the differences in the structure of the functional repertory according to water type (Fig. 2). The paragraph referred to in this comment has been deleted from the revised manuscript to prevent any confusion.

Comment: Line 278: I do not follow what is meant here, didn't the authors classify functional genes from shotgun sequencing?

Response: As mentioned in our response to the previous comment, this sentence has been removed from the revised manuscript.

Comment: Lines 301-306: Is it possible that some bacteria associated with FDOM are derived from terrestrial organic matter leaching into the riverine water?

Response: This is a very interesting remark. Indeed, some of the genera associated with FDOM are exclusively aquatic (e.g. *Polynucleobacter*), while others such as *Methylobacterium*

and *Acinetobacter* are often found in humic soils. The following sentence has been added to the revised Discussion (lines 357-360):

“ Interestingly, while Polynucleobacter is exclusively aquatic, Methylobacterium and Acinetobacter are often found in humic soils (65, 66) and could potentially be derived from terrestrial organic matter that leached into the riverine water.”

Comment: Lines 373-379: I don't think this is really needed in the conclusions, as this mostly repeats the methods.

Response: These sentences were removed from the revised manuscript.

Comment: Lines 383-384: Remove repetition of methods

Response: This sentence was removed from the revised manuscript.

Comment: Line 404: What is this Nucleic Acid Preservation buffer? Write the contents in the text. Will it keep RNA stable?

Response: NAP buffer does preserve the integrity of RNA. The following details have been added to the revised manuscript (lines 400-403):

“Immediately upon collection, filters were stored in 2 mL of Nucleic Acid Preservation (NAP) buffer. NAP buffer, which preserves DNA and RNA integrity at room temperature, contains EDTA disodium salt dihydrate, sodium citrate trisodium salt dihydrate and ammonium sulfate (71, 72).”

Comment: Line 449: Shouldn't there be a reference for EPA Method 160.2?

Response: The following reference was added to the text and to the reference list:

United-States Environmental Protection Agency. 1983. Methods for Chemical Analysis of Water and Wastes, EPA-600/4-79-020, USEPA, Method 160.2.

Comment: Line 489-496: Most journals accept supplementary data files and I think the authors should utilize this and not solely rely on an external service (which can be both edited and removed after publication). ASV tables, KEGG tables, scripts etc. can all be uploaded as supplementary files accompanying the manuscript.

Response: The script (16S_rRNA_analysis_script.txt), the ASV tables (ASV_table_bacterioplankton.csv) and KEGG table (KEGG_ALL.txt) were all uploaded as supplementary files in the revised submission. The following information was added to the “Data availability” section (lines 509-511):

“The main script for statistical analysis, the ASV tables and the main KEGG table used were all uploaded as supplementary material.”

Comment: Line 513: Details on the methods for multiplexing with indexes are missing. This information is needed to be able to reproduce the study.

Response: The following details on multiplexing were added to the revised manuscript (see lines 519-541):

“The first PCR was performed with primers specific to the V3-V4 region of the 16S rRNA gene (primers 347F and 803R) (80), which were tailed on the 5’ end with part of the Illumina TruSeq adaptors (Oligonucleotide sequences © 2007-2013 Illumina, Inc. All rights reserved). The following oligonucleotide sequences were used for amplification for the first PCR (actual primer sequence is in bold, the rest corresponds to the adapter sequence):

Forward primer (347F):

*5’-ACACTCTTTCCCTACACGACGCTCTTCCGATCT**GGAGGCAGCAGTRR**GGAAAT-3’,*

Reverse primer (803R):

*5’-GTGACTGGAGTTCAGACGTGTGCTCTTCCGATCT**CTACCRGGGTATCTAATCC**- 3’,*

Then, a second PCR was performed to attach remaining adaptor sequence (the regions that anneal to the flowcell and library specific barcodes).

Generic forward second-PCR primer: 5’-

AATGATACGGCGACCACCGAGATCTACAC[index1]ACACTCTTTCCCTACACGAC- 3’

Generic reverse second-PCR primer:

5’-CAAGCAGAAGACGGCATACGAGAT[index2]GTGACTGGAGTTCAGACGTGT-3’.”

Comment: Lines 517-18: There is no reason to keep the 16S bioinformatics in the supplementary methods. Move to the main text.

Response: All the section on 16S rRNA bioinformatics was moved to the main text (see lines 554-579).

Comment: Line 521: phyloseq package version?

Response: Phyloseq package version 1.32.0 was used. This information was added to the revised manuscript (see lines 576-579):

“An analysis of Shannon diversity according to sampling depth for each sample can be found in Suppl. Fig. 2, 4. ASV tables, metadata files and taxonomy information were incorporated into phyloseq objects (82) (phyloseq v. 1.32.0) before downstream analyzes.”

Comment: Line 523-530: The metagenome bioinformatics supplementary methods should be moved to the main text. Some brief details on the additional metagenome samples would also be useful, such as what type of waters were sampled (black, white, clear)? Was it surface or bottom water?

Response: The metagenome bioinformatics methods were moved to the main text of the revised manuscript (see lines 582-618). Details on these metagenomes have been provided by Santos *et al.* (2020) and this information was added to the revised manuscript (see lines 588-590):

“Details on the environments where these metagenomes were collected are provided as Supplementary data in Santos et al. (2020) (21).”

Comment: Lines 532-547: This section describes what statistical tools were used, but not how it was done. What R packages were used? Or other software? Was the data normalized (e.g. rarified) before alpha diversity analysis?

Response: The following information (**in bold**) was added to the revised manuscript (see lines 624-636):

*“ We first computed the Shannon H diversity index **using plot_richness from R phyloseq package** (Suppl. Fig. 3) and described the relative abundance of the different bacterial phyla in **stacked barplots from ggplot2** (Suppl. Fig. 1) in each water type. Then, we computed distance-based redundancy analyses (RDA) **using capscale from vegan R package** (84) to summarize the variation in the response variables (i.e. bacterioplankton communities) explained by environmental parameters (explanatory variables) (Fig. 2). Relative abundance (sum normalization) tables of ASVs were used as inputs for RDA analyses. We checked for multicollinearity by measuring variance inflation factors (VIF) **using vif.cca from vegan** (85), and then by computing stepwise variable selection **using ordistep from vegan** (86). The proportion of the variance explained by each variable was checked **using envfit, also from vegan** (86). Only environmental parameters with VIF < 10 and **selected by ordistep** were kept for RDAs (Fig. 2).”*

The data was not rarefied before alpha diversity analysis as rarefaction is not advisable when analyzing this type of dataset (McMurdie and Holmes 2014):

McMurdie PJ, Holmes S (2014) Waste Not, Want Not: Why Rarefying Microbiome Data Is Inadmissible. PLoS Comput Biol 10(4): e1003531. <https://doi.org/10.1371/journal.pcbi.1003531>

Comment: Line 536: Is it the Shannon's H diversity index?

Response: Yes, Shannon's H diversity index was used. The following information was added to the revised manuscript (lines 624-625):

“We first computed the Shannon H diversity index using plot_richness [...]”

Comment: Line 538-545: Was the RDA and NMDS based on ASV level or e.g. genus level? Was the data normalized before these analyses (e.g. relative abundances)?

Response: The RDA and NMDS were both based on a relative abundance table of ASV (a sum normalization was used). However, the NMDS analysis was removed from the revised Supplementary material, as it did not bring any new result that was not already shown in the RDA from Fig. 2 - the ordination analysis included in the main manuscript. The following information (**in bold**) was added to the revised manuscript (lines 627-631):

*“Then, we computed distance-based redundancy analyses (RDA) using capscale from vegan R package (84) to summarize the variation in the response variables (i.e. bacterioplankton communities) explained by environmental parameters (explanatory variables) (Fig. 2). **Relative abundance (sum normalization) tables of ASVs were used as inputs for RDA analyses.**”*

Comment: Line 546: How come only unweighted unifrac distances were used? This will only consider presence and absence information, and not the relative abundance of the ASVs.

Response: As mentioned in the response above, the NMDS analysis where Unifrac was used was removed from the revised Supplementary material, as it did not bring any new result that was not already shown in the RDA of Fig. 2, which provide a very similar (and more accurate) ordination approach and is already included in the main manuscript text.

Comment: Lines 572: Should it be > 0.5 and < -0.5 ?

Response: In this sentence, only > 0.5 is used as no significant negative correlation with Spearman correlation value < -0.5 was detected in this analysis.

Comment: Lines 572-576: Was any add-on used with Cytoscape? E.g. CoNet?

Response: No add-on was used on Cytoscape, the standard software was used.

Comment: Lines 585: I think "We thank two anonymous reviewers for their essential input on the original version of this manuscript." is no longer relevant in the acknowledgements.

Response: This sentence was removed from the revised manuscript.

Comment: Table 2: Here it would be useful to also include a column of "Water color" as in Table 1.

Response: A column of "Water color" was added in Table 2 as suggested.

Comment: Figure 1: Shouldn't this caption have at least one reference where the information was retrieved from?

Response: The following two references were added to the caption of Figure 1.

de Gonzalo G, Colpa DI, Habib MHM, Fraaije MW. 2016. Bacterial enzymes involved in lignin degradation. J Biotechnol, 236: 110-119.

Kamimura N, Takahashi K, Mori K, Araki T, Fujita M, Higuchi Y, Masai E. 2017. Bacterial catabolism of lignin-derived aromatics: New findings in a recent decade: Update on bacterial lignin catabolism. Env Microbiol Rep, 9: 679-705.

The caption of Figure 1 of the revised manuscript has been modified to the following (lines 972-973):

"Figure 1: The main steps of biological degradation of lignin. Humic DOM (in red) is produced after the initial oxidation of lignin-derived compounds (58, 68)."

Comment: Figure 1: Oxidation is a French word. Should it be Oxidation?

Response: The word has been corrected to "Oxidation" in the revised Figure 1.

Comment: Figure 2: "inferred functional repertory". Is this based on the functional annotation of the metagenomes or the taxonomy of the amplicon/metagenome data?

Response: The “inferred functional repertory” referred to here is based on the taxonomy of the 16S rRNA amplicon data (from which we inferred the functional repertory based on the bacterial functions detected in the shotgun metagenomes from the literature). The following information (in bold) has been added to the revised caption of Figure 2 (lines 975-980):

*“Figure 2: The of global and transcriptionally-active bacterioplankton communities, and their functional profiles significantly cluster according to water type. RDA ordination plots of sampling sites according to the (a) global bacterioplankton taxonomic structure, (b) transcriptionally-active bacterioplankton taxonomic structure and (c) inferred functional repertory. **The inferred functional repertory is based on the taxonomy of the 16S rRNA amplicon data.**”*

Comment: Figure 3: The relative abundance scale does not have any numbers. How much does the scale represent, is it 0-100%?

Response: The relative abundance scale does not have any numbers in Figure 3 because the color intensity displayed in the heatmap does not represent a fixed range of abundance values. Instead, the lightness/darkness of each ASV x sample combination is scaled according to the abundance of the same ASV in all other samples. Thus, since the relative abundance is relative to what is found in other samples, and does not depend on the abundance of different ASVs from the same sample, it is impossible to establish a common scale with abundance values for all ASVs (all ASVs would need their own scale). The following information has been added to the revised caption of Figure 3 (lines 994-995):

“The relative abundance (color intensity) of each ASV in each sample is scaled according to the abundance of the same ASV in all other samples.”

Comment: Figure 5: I think that the correlation network is not very informative. Typically each line will also show if it's a positive or negative correlation, here instead all lines are grey or colored based on taxonomy. Some lines are also black and I am not sure what this color denotes. The taxa is also only shown on a high taxonomic level.

Response: No significant negative correlation with a Spearman correlation value < -0.5 was detected in this analysis, thus only positive correlations are shown in Figure 5. Black edges represent direct correlations between humic FDOM and specific ASVs, while grey edges represent indirect correlations. The following information is mentioned in the caption of Figure 5 (lines 1018-1020):

“In (c,d) black edges represented direct correlations between humic FDOM and specific ASVs, while grey edges represent indirect correlations. All correlations in (c,d) are positive correlations.”

The ASVs from this figure are shown at the best taxonomic level possible.

Comment: Figure 6: The color scale has no numbers and is only indicated by minus and plus symbols. What is yellow and red? Is it -1 to 1 Spearman's rho?

Response: The color scale does not have any numbers in Figure 6 because the color intensity displayed in the heatmap does not represent a fixed range of correlation values. Instead, the color intensity of each genus x pathway combination is scaled according to the correlation values of the same pathway in all other genera. Thus, since the color is relative to what is found

in other genera, and is not directly associated to the Spearman correlation values, it is impossible to establish a common scale with correlation values for all pathways (all pathways would need their own scale).

Comment: Figure 7: This should be figure 1

Response: The figure numbers have been carefully reviewed in the revised manuscript.

Comments on the supplementary methods-----

Comment: What settings were used with the different DADA2 steps, such as mergePairs?. For example, was default settings used? This information will make it possible to reproduce the study.

Response: When not specified in the text, the default settings were used. However, to prevent any confusion, we added the following specifications (**in bold**) in the revised methods (lines 556-565):

*“Quality control of reads was done with the filterAndTrim function using the following parameters: 290 for the forward read truncation length, 270 for the reverse read truncation length, 2 as the phred score threshold for total read removal, and a maximum expected error of 2 for forward reads and 3 for reverse reads. The filtered reads were then fed to the error rate learning, dereplication, and ASV inference steps (**with default settings**) using the functions learnErrors, derepFastq, and DADA, which are all from the DADA2 pipeline (81). The merging of sequence pairs was done using mergePairs (**with default settings**) also from DADA2. Chimeric sequences were removed using the removeBimeraDenovo function (**default settings**) with the “consensus” method parameter.”*

Comment: What settings was used with blastn?

Response: The following information (**in bold**) was added to the revised manuscript (lines 573-575):

*“Sequences with no matches above the identity threshold were assigned taxonomy using a lowest common ancestor method generated on the top 50 matches using blastn (**e-value cutoff of 1e-4 with default parameters**).”*

Comment: What was the date of the ncbi database?

Response: The following information (**in bold**) was added to the revised manuscript (lines 567-569):

*“Taxonomic annotation of amplicon sequence variants (ASV) was performed by using blastn matches against NCBI “16S Refseq Nucleotide” database (**11/2020**).”*

Comment: What is the 16S Microbial database exactly?

Response: The 16S Microbial database refers to the “NCBI 16S Refseq Nucleotide database”. The name has been clarified in the revised manuscript (line 567-569):

“Taxonomic annotation of amplicon sequence variants (ASV) was performed by using blastn matches against NCBI “16S Refseq Nucleotide” database (11/2020).”

The database can be accessed on the following link:

https://www.ncbi.nlm.nih.gov/refseq/targetedloci/16S_process/

More information can be found here:

<https://www.ncbi.nlm.nih.gov/refseq/targetedloci/>

Comment: The authors write: "This method is closely inspired from the LCA algorithm implemented in MEGAN (Huson et al. 2016)." But they never explain what was done. It seems this method had a 99% identity threshold and might have caused many classifications to be ranked on a higher taxonomic level, with many ASVs also being classified as "Bacteria". I suggest to also classify the ASVs sequences against SILVA and use this as a comparison.

Response: Please refer to prior justification for not using DADA2 methodology.

Comment: What trimmomatic data was used? Was it the filtered paired, filtered unpaired, or both? This information is required to be able to reproduce the study.

Response: Since all samples were merged in a co-assembly process, unpaired filtered reads from trimmomatic were used in order to maximise diversity and coverage. The following information (in **bold**) was added to the revised manuscript (lines 592-593):

*“Trimmomatic **unpaired** filtered reads (default parameters) were assembled using the megahit assembler [...].”*

Comment: Was each sample assembled using megahit individually or was a co-assembly conducted? This needs to be clarified.

Response: All samples were used in a co-assembly in order to maximise diversity and coverage. The following information (in **bold**) was added to the revised manuscript (lines 592-593):

*“Trimmomatic unpaired filtered reads (default parameters) were assembled using the megahit assembler **in a co-assembly** [...].”*

Comment: The authors mention that a table comprising the abundance of pathways in each sample was produced. However, the authors mention no methods on mapping reads onto the assembly so how did the authors get information on gene abundances?

Response: Abundances were inferred from 16S rRNA counts. The 16S counts were directly correlated to the relative abundance of the different bacterial taxa, and we inferred the relative abundance of bacterial functions (i.e. the functional repertory) through it.

Reviewer #2

Comment: Sylvain et al. investigated the bacterioplankton communities in Amazonian black water. Given the importance of Amazonian River in global ecosystem services, I believe the data presented in the current study are valuable. Since this is an easy study, I do not have many concerns. All displays are eye-catching.

Response: Thank you for these comments!

However, I think the paper is not well organized and written. I would suggest the readers to read more related papers and re-organize this paper. Below I listed some examples.

Response: We carefully revised the paper according to the suggestions of both reviewers and to recently published literature. In the revised manuscript, the Results are cited more clearly, the Discussion has been shortened and the overall flow of the paper has been improved.

Comment: In the beginning of the Abstract, I see no knowledge gap. Why this study should be conducted? Just because some phenomenon remain unsolved? Billions of things remain unsolved, but we needn't, and it's impossible, to investigate every UNSOLVED thing.

Response: The Abstract of the revised manuscript has been restructured. The knowledge gap (in **bold**) has been clarified to the following (see lines 22-28):

*“ The Amazon River basin sustains dramatic hydrochemical gradients defined by three water types: White, clear and black waters. In black water, important loads of allochthonous humic dissolved organic matter (DOM) result from the bacterioplankton degradation of plant lignin. **However, the bacterial taxa involved in this process are still unknown, since Amazonian bacterioplanktonic communities have been poorly studied. Their characterization could lead to a better understanding of the carbon cycle in one of the Earth’s most productive hydrological system.**”*

Comment: Line 39-41: This statement is too absolute. First, the MAGs were assembled from the literature, not you REAL data. Second, they may have these genes, but having genes does not necessarily function.

Response: This sentence has been modified to accurately reflect our results and to ensure the statement was less absolute (lines 38-41):

“ The strongest correlations were found in Polynucleobacter, Methylobacterium and Acinetobacter genera, three low abundant but omnipresent taxa which possessed several genes involved in the main steps of the β -aryl ether enzymatic degradation pathway of diaryl humic DOM residues.”

Comment: The authors gave two hypotheses in Introduction. Hypothesis 1 looks fine, but I would suggest changing it like 'Given that bacterioplankton communities have been demonstrated to involve in the transformation of DOM, we hypothesize that...'. If you propose this hypothesis just because 'as shown in other ecosystems', I may disagree because your ecosystem looks highly different from others.

Response: This sentence was modified according to the suggestion of the reviewer (lines 115-118):

“Given that bacterioplankton communities have been shown to be involved in the transformation of DOM (30-32), we hypothesized that the optical characteristics of DOM in particular FDOM, such as humic-like materials, would be one of the most important drivers of bacterioplankton communities.”

Comment: Line 81: A little confusing to me. Now that Rio Negro contains high DOM, why is it oligotrophic?

Response: The Rio Negro is an oligotrophic ecosystem (Almeida *et al.* 2018; Beltrão *et al.* 2019) as it presents hostile physicochemical conditions (extremely poor in nutrients and ions, and acidic pH) which leads to generally low aquatic productivity. The DOM contents in the Rio Negro are mostly allochthonous (terrestrial origin, from decomposing wood and leaves on the river bed), and not from autochthonous ecosystem productivity *per se*.

References:

Almeida FF, Santos-Silva EN, Ector L, Wetzel CE. 2018. *Eunotia amazonica* sp. nov. (Bacillariophyta), a common stalk-forming species from the Rio Negro basin (Brazilian Amazon). *European Journal of Phycology*. 53(2). doi: 10.1080/09670262.2017.1702372

Beltrão H, Zuanon J, Ferreira E. 2019. Checklist of the ichthyofauna of the Rio Negro basin in the Brazilian Amazon. *Zookeys*. 881:53-89. doi: 10.3897/zookeys.881.32055.

Comment: I have to say that I feel very tired to read the Results part, although, at I stated above, this is a very easy and small study. Let's look at the first paragraph of Results. The first sentence, it's not necessary to say 'The Amazonian bacterioplankton was similar in composition to what has been reported in (32)'. You just show your results, and discuss and compare your results with others in Discussion. Why start discussing in the first sentence of Results!

Response: This sentence was removed from the Results of the revised manuscript.

Comment: In Line 126, the authors stated 'While Shannon alpha diversity seemed largely unaffected by water colors'. Is this really valid result? What valid information can we get from this? Why not show your data?

Response: This sentence was modified to be more explicit about the results observed (lines 137-139):

“While Shannon alpha diversity did not significantly differ between water types ($p > 0.05$, mean Shannon diversity between 6-7 for all water types, see Suppl. Fig. 2, 3, 4), [...]”

The detailed results can also be seen on Supplementary Figures 2 and 5.

Comment: In Objective #1, the authors aimed to identify the environmental variables shaping the composition, transcriptional activity and inferred functions of Amazonian bacterioplankton. But in the main text, I don't see any specific variables. So, the authors, actually, failed to tell us anything in this long paragraph. WHY NOT SHOW YOUR DATA IN RESULTS? This is an important problem across the whole paper.

Response: The paragraph referred to in this comment has been rewritten to explicitly state the

environmental variables identified as bacterioplankton community drivers. The results are also detailed in Figure 2. See lines 152-168:

“ The environmental parameters that significantly influenced the taxonomic structure of global bacterioplankton, of transcriptionally-active bacterioplankton, and the inferred functional repertory of bacterioplankton were not identical. The taxonomic structure of global bacterioplankton from black water was mostly driven by the concentration of Cd^{2+} , Co^{2+} , humic DOC, fluvic DOC, total DOC and levels of SAC340, but in white and clear waters it was associated to the concentration of Mg^{2+} , Na^+ , and the levels of pH, conductivity and Abs254/365. The taxonomic structure of transcriptionally-active bacterioplankton in black water was driven by the concentration of Co^{2+} , Pb^{2+} , and Cd^{2+} , but in white and clear waters the main drivers were the concentrations of Ca^{2+} , Mg^{2+} , K^+ , protein-like DOC, fluvic DOC and the level of Abs254/365. The inferred functional repertory from blackwater was associated by the concentrations of Fe^{3+} , Mn^{2+} and Cd^{2+} , while in white and clear waters it was affected by the concentration of Ca^{2+} , silicate and the levels of pH and Abs254/365. Overall, the taxonomic structures and inferred functional repertory were driven by several parameters associated with the relative abundance of the different FDOM components (i.e. the relative abundance of humic, fluvic and protein-like DOC, in addition to the SAC340 and Abs254/365 ratios), which appear in red in RDAs of Fig. 2.”

The Results section of the revised manuscript was also carefully reviewed to ensure that the data associated to each statement was appropriately stated, for all the analyses performed.

Comment: Line 195: I've never heard about 'Co-abundance network'. Is this a newly invented word?

Response: The term “co-abundance network” is the most appropriate for our analysis, as the networks from our study were based on the relative abundance of bacteria or humic DOC, rather than presence/absence data only (which we would refer to as “co-occurrence networks”). The term “co-abundance network” is common in the literature, a few examples in the papers below:

Chen, L., Collij, V., Jaeger, M. *et al.* Gut microbial **co-abundance networks** show specificity in inflammatory bowel disease and obesity. *Nat Commun* **11**, 4018 (2020). <https://doi.org/10.1038/s41467-020-17840-y>

Irannia, Z.B., Chen, T. TACO: Taxonomic prediction of unknown OTUs through OTU **co-abundance networks**. *Quant Biol* **4**, 149–158 (2016). <https://doi.org/10.1007/s40484-016-0073-2>

Comment: Please try to shorten it. It's too long and boring.

Response: The Results and Discussion parts have been shortened/restructured. We are confident that the flow of the revised manuscript has been improved and that these modifications will benefit the experience of the readers.

Comment: Line 278 : PiCRUST should be PICRUST

Response: The paragraph where PICRUST is mentioned has been deleted in the revised manuscript.

Comment: Line 552: ntree = 100 is not a commonly use parameter. We commonly use 500. Using 100 can save time but increase error. Your data are not very large, why use 100?

Response: The Random Forest tests were reconducted, as suggested, using ntree = 500. Overall, we obtained the same results: The 40 ASVs responsible for the most important mean decrease in GINI coefficient in global and transcriptionally-active bacterioplankton (Fig. 3) were the same whether ntree = 100 or ntree = 500 was used.

May 3, 2023

Mx. Francois-Etienne Sylvain
Universite Laval
Biology
1030 avenue de la Médecine
Quebec, QC
Canada

Re: Spectrum04793-22R1 (Bacterioplankton communities in dissolved organic carbon-rich Amazonian black water.)

Dear Mx. Francois-Etienne Sylvain:

Thank you for submitting your manuscript to Microbiology Spectrum. Your paper is very close to acceptance. Please add a final sentence to your abstract concluding on the results and what they mean in a broader context. As this revision is quite minor, I expect that you should be able to turn in the revised paper in less than 30 days, if not sooner.

When submitting the revised version of your paper, please provide (1) point-by-point responses to the issues raised by the reviewers as file type "Response to Reviewers," not in your cover letter, and (2) a PDF file that indicates the changes from the original submission (by highlighting or underlining the changes) as file type "Marked Up Manuscript - For Review Only". Please use this link to submit your revised manuscript. Detailed instructions on submitting your revised paper are below.

Link Not Available

Sincerely,

Eva Sonnenschein

Reviewer comments:

Preparing Revision Guidelines

For complete guidelines on revision requirements, please see the journal Submission and Review Process requirements at <https://journals.asm.org/journal/Spectrum/submission-review-process>. **Submissions of a paper that does not conform to**

Microbiology Spectrum guidelines will delay acceptance of your manuscript. "

Please return the manuscript within 60 days; if you cannot complete the modification within this time period, please contact me. If you do not wish to modify the manuscript and prefer to submit it to another journal, please notify me of your decision immediately so that the manuscript may be formally withdrawn from consideration by Microbiology Spectrum.

Response to the Editor and to the Reviewers - Spectrum04793-22R2

Comment: Please add a final sentence to your abstract concluding on the results and what they mean in a broader context.

Response: The following sentence was added at the end of the Abstract (lines 42-44):

“ Overall, this study identified key taxa with DOM-degradation genomic potential, which involvement in allochthonous Amazonian carbon transformation and sequestration merits further investigation. ”

May 4, 2023

Mx. Francois-Etienne Sylvain
Universite Laval
Biology
1030 avenue de la Médecine
Quebec, QC
Canada

Re: Spectrum04793-22R2 (Bacterioplankton communities in dissolved organic carbon-rich Amazonian black water.)

Dear Mx. Francois-Etienne Sylvain:

Your manuscript has been accepted, and I am forwarding it to the ASM Journals Department for publication. You will be notified when your proofs are ready to be viewed.

Sincerely,

Eva Sonnenschein
Editor, Microbiology Spectrum
